



# On the significance of rain droplet slowdown and deformation for leading-edge rain erosion

Nils Barfknecht[1] and Dominic von Terzi[1]

[1]Wind Energy Group, Faculty of Aerospace Engineering, Delft University of Technology, The Netherlands

**Correspondence:** Nils Barfknecht (n.barfknecht@tudelft.nl)

**Abstract.** Leading-edge rain erosion is a severe problem in the wind energy community since it leads to blade damage and a reduction in annual energy production in the order of several percent. The impact speed of rain droplets is a key driver for the erosion rate; therefore, its precise computation is essential. This study investigates the aerodynamic interaction of rain droplets and wind turbine blades. Based on findings from the literature and an analysis of the relevant parameter space, it is
found that the aerodynamic interaction leads to a reduction in the impact speed. Additionally, the rain droplets deform and break up as they approach the wind turbine blade. An existing Lagrangian particle model, developed for research in aircraft icing, is adapted, extended, and validated for leading edge rain erosion to study the process in more detail. Results show that the droplet slowdown reduces predicted damage toward the tip of the blade by over 50 %. The model indicates that the aerodynamic blade interaction affects small droplets significantly more than large droplets. Due to this drop size dependency,
the damage accumulation is shifted towards higher rain intensity events. Additionally, the droplet impact speed is sensitive to the aerodynamic nose radius of the airfoil. Due to this sensitivity and its drop size dependency, the slowdown effect provides interesting levers for erosion mitigation via blade design or operational adjustments. To conclude, the aerodynamic interaction between droplet and blade is non-negligible and needs to be taken into account in erosion lifetime models.

## 1 Introduction

Leading-edge rain erosion is a severe problem for wind turbines. During precipitation events, hydrometeors impact the blade and, over time, create roughening. This damage can grow to large pits that can reach deep into the different structural layers of a wind turbine blade. These damages must be repaired periodically to prevent wind turbine blade failure. Roughening also disturbs the airflow over the blade and leads to a loss in annual energy production (AEP) (Barfknecht et al., 2022). Forecasting and understanding the mechanisms of erosion is important for maintenance, but also for operational adjustments of the turbine,
such as the erosion safe mode (Bech et al., 2018).

A key parameter that influences the rain erosion lifetime is the impact speed of the rain droplets. A common approach is to relate the droplet impact speed $V_{\text{impact}}$ via a power law to an incubation metric $N$. $N$ is a measure for the incubation time, which is the operational time until visible erosion damage occurs:

$$N \propto \frac{1}{V_{\text{impact}}^{\beta}},\tag{1}$$





where $\beta$ is a constant. $N$ can have various meanings depending on the damage model, such as the number of impacts or the impingement. Common to all models is that the magnitude of the parameter $\beta$ is significant. Parameters for $\beta$ reported in literature are 5.7 in Hoksbergen et al. (2022), 16.92 in Shankar Verma et al. (2021) and 7.2-10.5 in Bech et al. (2022). While the reported values differ significantly based on the test apparatus used and exact definition of $N$, they all preserve the character of the equation, namely that small changes in $V_{\text{impact}}$ will yield vastly different erosion lifetimes. It is, therefore, important

to accurately determine the impact speed. The impact speed is the surface normal component of the impact vector, which is calculated as the difference between the blade and droplet velocity vector. The droplet's velocity vector is usually considered to be comprised of components of the droplet's terminal velocity and its advection velocity with the wind (Barfknecht et al., 2022; Verma et al., 2020). However, in aeronautics, it has been known already for a long time that rain droplets and wings can interact aerodynamically (Nicholson, 1968). This leads to rain droplet deformation and slowdown when observed from

the wing (Vargas and Feo, 2011). Thus adding an extra velocity component to the problem. The potential slowdown of rain droplets has so far received no attention in the wind energy community. One exception is Prieto and Karlsson (2021), where, however, only limited results for spherical droplets were obtained. No droplet deformation was included in their analysis.

Sor et al. (2019) performed measurements in which water droplets were seeded in a rotating-arm test-rig. A blunt airfoil was mounted on the arm. High-speed photographs were taken that showed the droplets close to impact with the wing. Figure 1

shows an excerpt of their results. As the airfoil approaches the droplets, the droplets start to deform from a spherical to an oblate shape. Shortly before impact, the droplets undergo breakup. While the experiments were performed for aircraft icing research, the parameter space fits the one encountered in leading-edge erosion of wind turbines very well. These findings stand contrary to current practice in leading-edge erosion research, where it is assumed that the droplets are spherical at impact (Hoksbergen et al., 2023; Fæster et al., 2021; Keegan et al., 2012; Verma et al., 2020). The measurements of Sor et al. (2019) imply that rain

droplets can undergo breakup, and, therefore, the rain droplets' appearance at impact can be considered complex in shape.





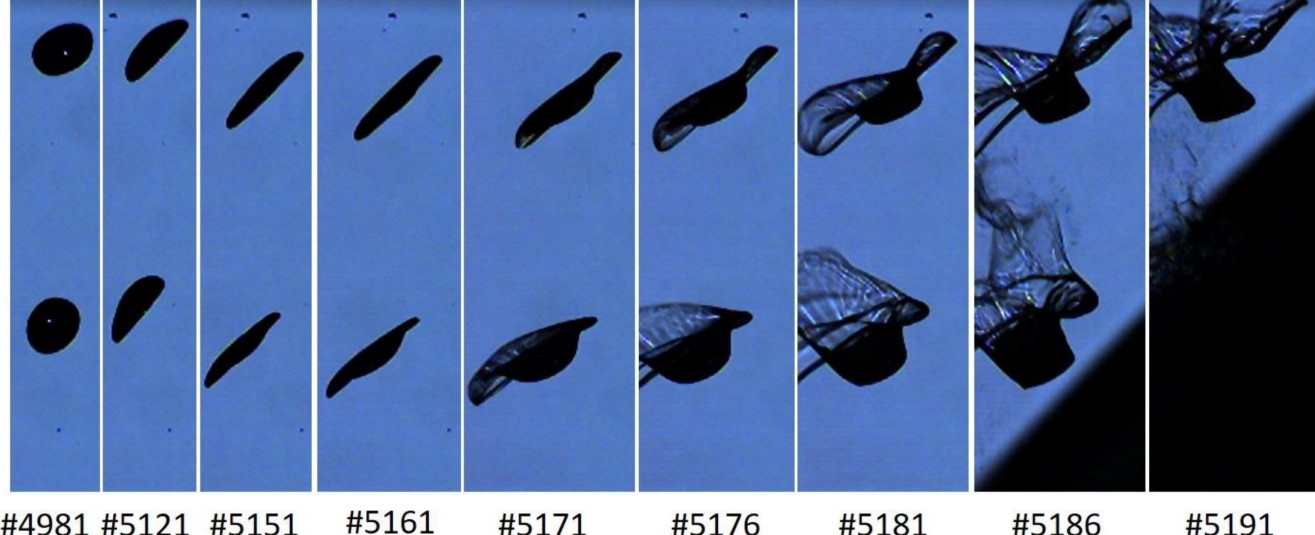

#4981 #5121 #5151 #5161 #5171 #5176 #5181 #5186 #5191

**Figure 1.** High-speed photography of falling water droplets of 1.75 mm diameter approaching an airfoil in a rotating-arm test-rig. The different frames show the temporal progression; the shadow that can be observed in the last two frames is the approaching airfoil; the droplets are first round, then become oblate and, before impact, break up with smaller droplets being emitted from the rim; free-stream velocity of 60 m s$^{-1}$, airfoil chord of 1.05 m; the photographs are reproduced from Sor et al. (2019).

Figure 2 shows the results of a similar experiment performed by Vargas and Feo (2011). It can be observed how the rain droplets' speed changes in front of the airfoil. The water droplets slow down as they approach the leading edge of the rotating airfoil. Droplets with a free-stream velocity of 90 m s$^{-1}$ experience a velocity reduction of almost 12 m s$^{-1}$. Considering the exponent of the damage law in Eq. 1, this effect is highly relevant. It appears, therefore, that the effect of droplet slowdown and deformation cannot be neglected when studying leading-edge erosion by rain and needs to be further understood.

The research presented here investigates the impact of rain droplet slowdown and deformation on the erosion lifetime prediction of wind turbine blades. It is important to note that this study assumes that the problem is observed in the reference frame of the airfoil. From an airfoil's perspective, the incoming droplet's speed reduces; hence the term *slowdown* is used. An observer located on the ground will see the droplets gain speed. Since the effect reduces the impact speed, the term slowdown seems appropriate. The paper is organized as follows: Sect. 2 presents the methodology. This includes a brief introduction to droplet deformation and breakup. Additionally, the parameter space of the problem is investigated. Based on these findings, an existing droplet model, developed for research in aircraft icing, is adapted, extended, calibrated, and validated to study the slowdown and deformation process. Subsequently, Computational Fluid Dynamics (CFD) simulations are performed on airfoils of reference turbines to obtain their background velocity field to determine parameters needed in the model. Finally, the used precipitation data and the damage model are discussed in further detail. In Sect. 3, our proposed slowdown and deformation model is employed to analyze the sensitivity of the droplet model with respect to the droplet diameter and the airfoil's aerodynamic nose radius. This is followed by combining the model with the precipitation data and then computing the impact

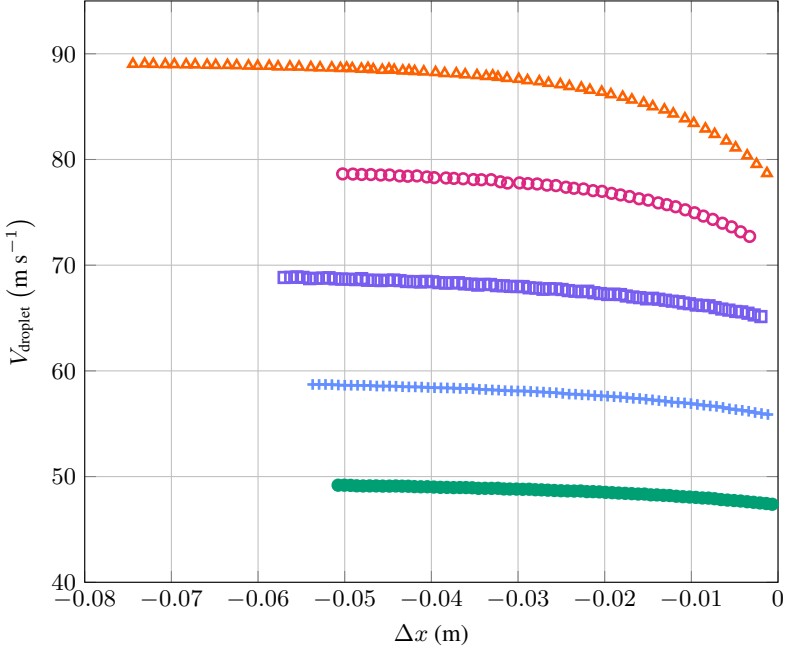

**Figure 2.** Measured velocity of a 0.49 mm diameter water droplet as a function of the distance to the leading edge of an airfoil. Blunt airfoil with a chord length of 0.47 m; five different free-stream velocities of: 50 m s$^{-1}$: ●, 60 m s$^{-1}$: +, 70 m s$^{-1}$: ▫, 80 m s$^{-1}$: ○ and 90 m s$^{-1}$: △; velocities are observed from the airfoil; data were collected in a rotating-arm test-rig and are reproduced from Vargas and Feo (2011).

of the droplet slowdown and deformation on the lifetime of two model turbines. Finally, in Sect. 4 a summary is provided, conclusions are drawn and recommendations are given.

## 2 Methodology

### 2.1 Discussion of the underlying physics

Before a suitable approach can be chosen to model the droplet slowdown and deformation, an understanding of the physics of the droplet along its trajectory is necessary. A complete review of the known processes encountered during aerodynamic droplet deformation and breakup is out of the scope of this work. However, since droplet deformation and breakup is a rather new phenomenon for the leading edge erosion community, a brief summary with a discussion of the parameter space seems appropriate.

For aerodynamic droplet deformation and breakup, the important non-dimensional numbers are the Weber number (We) and the Ohnesorge (Oh) number (Jackiw and Ashgriz, 2021). They read

$$\text{We} = \frac{\rho_{\text{air}} V_{\text{slip}}^2 \phi_0}{\sigma_{\text{water}}}, \qquad\qquad \text{Oh} = \frac{\mu_{\text{water}}}{\sqrt{\rho_{\text{water}} \sigma_{\text{water}} \phi_0}}, \qquad\qquad (2)$$

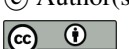



with density $\rho$, surface tension $\sigma$ and dynamic viscosity $\mu$, where the subscripts $_{air}$ and $_{water}$ indicate the corresponding medium. $\phi_0$ represents the droplet diameter and $V_{slip}$ is the slip velocity, i.e. the difference between the velocity of the air ($V_{air}$, see Section 2.3) and the drop ($V_x$, see Sect. 2.2). The Weber number relates the inertial forces to the surface tension forces, whereas the Ohnesorge number relates the viscous to the inertial and surface tension forces. Depending on the Weber number, droplets subject to aerodynamic forces can first undergo deformation and subsequently also break up. Figure 3 shows an often-cited

graph taken from Hsiang and Faeth (1995). It depicts how droplets are expected to behave depending on the Weber and Ohnesorge number. From the figure, it is evident that for Oh $< 0.1$, the expected behavior is a function of the Weber number only.

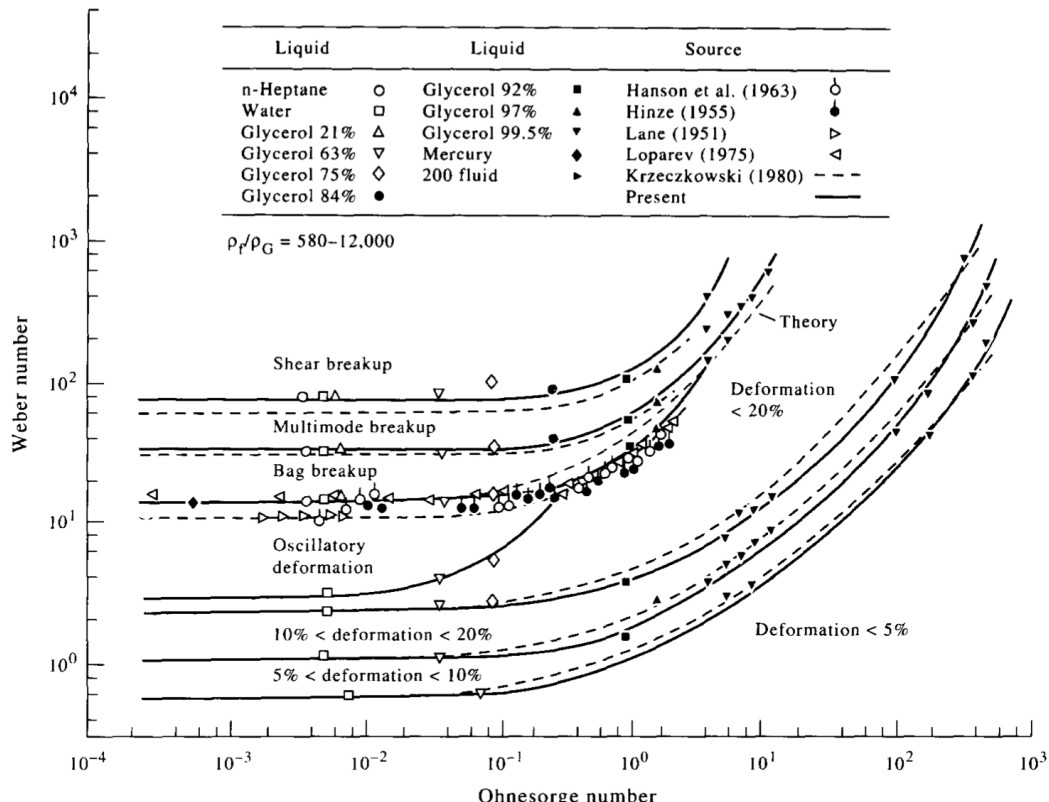

**Figure 3.** Droplet deformation and breakup modes as a function of the Weber and the Ohnesorge number; the figure is reproduced from Hsiang and Faeth (1995).

Aerodynamic droplet breakup consists of two phases, the initiation, also called the deformation phase, and the breakup phase (Jackiw and Ashgriz, 2021). During the deformation phase, the droplet's shapes flattens. At some point the droplet breaks up

into smaller droplets. This is process is also called secondary breakup. Different breakup modes exist such as bag, bag and stamen, multimode and shear breakup. Some of these modes are shown in Table 1. After the breakup stage is complete, the original droplet will have decayed into a series of small drops that can be characterized by a drop size distribution. Subsequently,





the resultant droplets might deform and break up again, forming a decay cascade. For more information about the fundamental mechanics of droplet dynamics, the reader is recommended to read (Jackiw and Ashgriz, 2021, 2022).

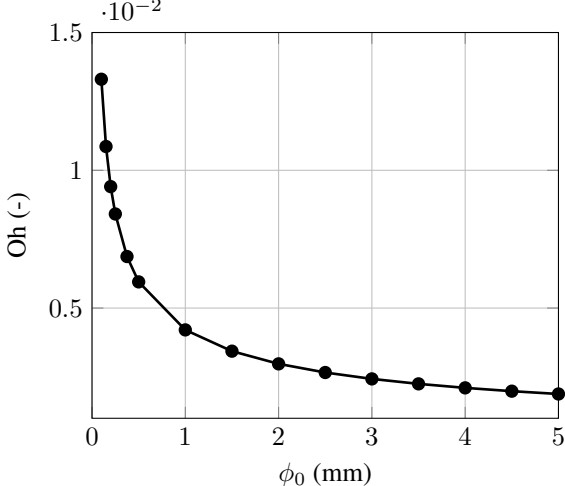

**Figure 4.** Ohnesorge number plotted against the rain droplet diameter; the values of the Ohnesorge stay below 0.1.

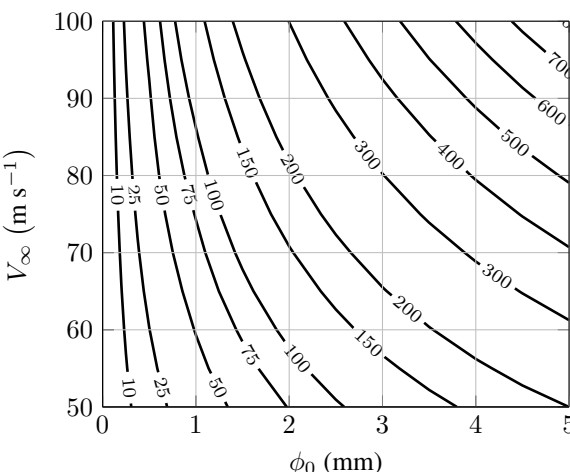

**Figure 5.** Contour plot of Weber numbers at impact time as a function of droplet diameter and free-stream velocity; the slip velocity required for the Weber number computation was calculated with the model from Sect. 2.2; the model parameters are $R_c = 0.07$ m and $n = 1.1$.

Two figures were created to analyze the parameter space for the leading-edge erosion problem in more detail. Figure 4 and 5 show the Ohnesorge and the Weber numbers, respectively, for droplets of varying sizes and free-stream velocities.[1] This set of simulations spans the parameter space in terms of non-dimensional numbers for the erosion problem. The Ohnesorge number is not dependent on the flow velocity and is, apart from the physical constants, a function of the droplet diameter only. Figure 4 shows that the Ohnesorge numbers stay below 0.1, indicating that the droplet breakup is governed by the Weber number only.

The Weber numbers lie in a very broad range of 1 to 800. The wide range of Weber numbers encountered in this problem leads to very different droplet behaviors. The droplet behavior is expected to range from simple deformation for small and slow droplets to shear breakup for larger and faster droplets. In Table 1, example images of the different breakup modes in a rotating-arm test-rig are given, together with an approximate Weber number close to impact.

    Most fundamental research in the literature about droplets is based on experiments in shock tubes and steady disturbances

(Hsiang and Faeth, 1995). However, in the present problem, the droplets traverse through a velocity field that changes depending on the distance to the airfoil. Figure 6 gives an example velocity field. Therefore, it is impossible to directly translate the graph of Fig. 3 to, e.g., the outcomes in Table 1. For this problem, the shape and extent of the background velocity field must also be considered. It is intuitive to assume that a larger airfoil will have more influence on the behavior of the droplet than a

---

[1]As will be shown later in Fig. 21 rainfall is almost exclusively composed of droplets in the range from 0 to 4 mm size. Some instances of larger droplets have been recorded in the literature (Jones et al., 2010).




**Table 1.** Examples of droplet deformation and breakup in the measurement campaigns of García-Magariño (2016); estimated Weber numbers at impact calculated with the model from Sect. 2.2.

| Image | Estimated We | Mode | $\phi_0$ (mm) | $V_\infty$ (m s$^{-1}$) | Reference in García-Magariño (2016) |
|---|---|---|---|---|---|
| 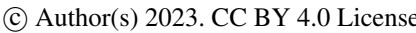 | $\approx 29$ | Deformation | 0.788 | 50 | Fig. 3.8 |
| | $\approx 17$ | Bag | 0.191 | 90 | Fig. 5.6 |
| | $\approx 72$ | Bag and stamen | 0.782 | 90 | Fig. 3.7 |
| | $\approx 388$ | Shear | 3.2 | 90 | Fig. 5.9 |

small airfoil, even though the Weber number of the droplet is similar for both airfoils close to impact. Therefore, one needs to
conclude that, while the general body of droplet breakup and deformation is extensive, only very limited knowledge exists that
is applicable to the wind turbine rain erosion problem.

Since current erosion research treats droplets as spherical and thus as a coherent structure when impacting with a blade, it is
also assumed that the entire water mass of a single droplet possesses the same impact velocity. The question is whether droplet
breakup invalidates this assumption. If the velocity that describes the maximum extent of the droplet's water mass is in the
same order as the droplet slowdown itself, then, with Eq. 1 in mind, the damage potential of a droplet might be significantly
influenced. To understand this aspect further, additional frames of the 0.191 mm droplet from Table 1 are shown in Fig. 7. In six
distinct frames, a purple and an orange arrow indicate the maximum extent of the bag that forms during the breakup. With the
timestamp and indicated length scale, the growth velocity of this bag can be obtained by using a simple Backward Euler Finite
Difference scheme. The obtained velocities are indicated next to the frame number in the figure. It can be seen that the velocity
is fairly low in Frame 2 and 3 when the bag is just beginning to form. However, as soon as the bubble starts rapidly growing,

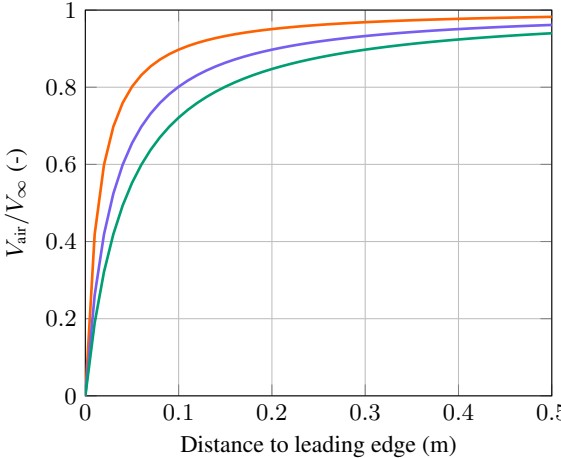

**Figure 6.** Non-dimensional velocity field along stagnation streamline vs. the dimensional distance to the leading edge; the FFA-W3-211 airfoil of the IEA 15MW turbine at 0-degree angle of attack was chosen; 0.5 m chord: ——, 1.0 m chord: ——, 1.5 m chord: ——.

the velocity quickly increases to a peak of 42.66 m s$^{-1}$. Close to impact, this velocity reduces to a still significant value of 22 m s$^{-1}$. This example shows that the water's velocity inside a droplet that undergoes breakup (close to impact) is not constant in space and time. The exact velocity distribution inside the droplet is probably breakup-mode dependent, and droplets that only undergo gradual deformation will preserve a reasonably constant velocity throughout the droplet. To further elaborate on this argument, if droplets fracture into sub-droplets during a breakup, all resulting droplets will have a distinct impact velocity. To conclude the findings, experiments suggest that droplets approaching wind turbine-sized airfoils are either deformed or will show breakup shortly before impact. Additionally, droplets that undergo breakup will show a non-homogeneous impact velocity distribution across their water mass.



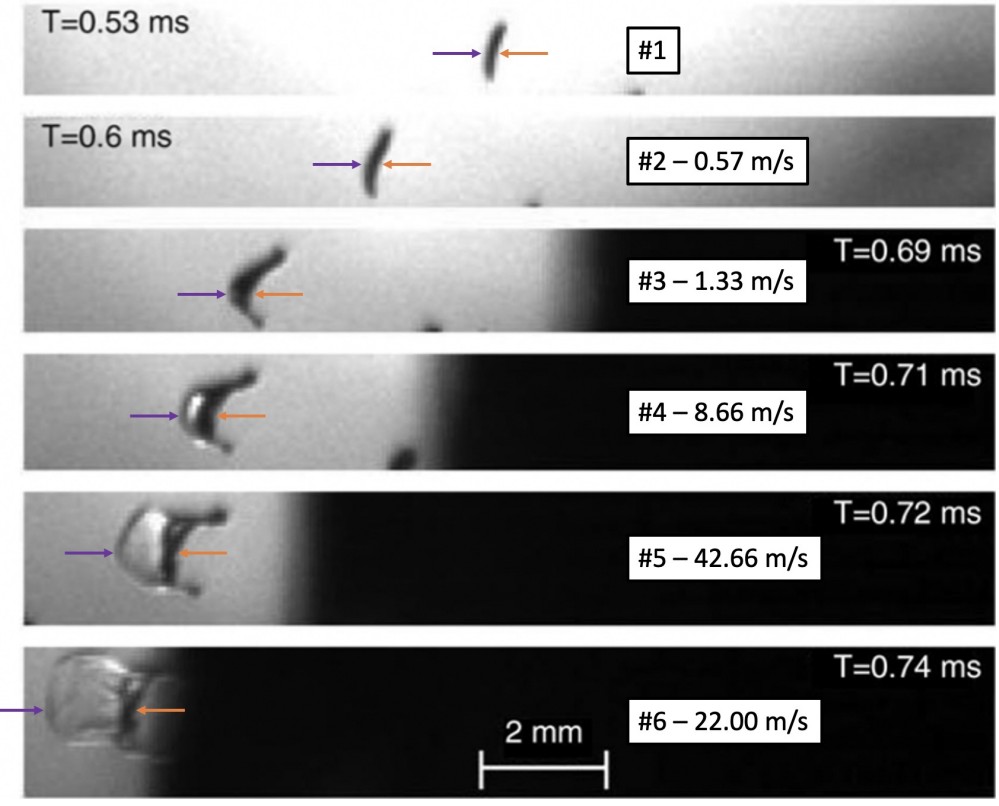

**Figure 7.** 0.191 mm diameter droplet approaching an airfoil and showing a bag breakup mode; the scale, time, frame identifier, and bag expansion speed are also indicated; the photographs are reproduced from Fig. 5.6 of García-Magariño (2016).

## 2.2 One-way coupled Lagrangian particle model

The influence of droplet deformation and breakup on the blade lifetime under erosion is investigated with a model that adequately describes the relevant physical processes. Various Lagrangian droplet deformation models exist in the literature, such as the TAB, NLTAB3, DDB, and DRD models (Sor and García-Magariño, 2015; Schmehl, 2004). However, to the author's knowledge, to date, no single Lagrangian model can describe the full range of complex phenomena of droplet slowdown and breakup in sufficient detail. Some advanced models attempt to model particular regimes, such as in Sichani and Emami

(2015) for a droplet under deformation and up to the onset of bag rupture. Direct numerical simulation (DNS) could capture all relevant physics and phenomena, especially when higher-order numerical schemes are applied. However, its computational expense makes it prohibitive when a large parameter space is supposed to be studied. Thus a gap exists with computationally affordable but low-accuracy Lagrangian particle models on one side and highly accurate but extremely costly DNS codes on the other.

This dilemma is resolved by simplifying the problem based on educated assumptions. In particular, it is argued that the model's foremost aim must be the accurate prediction of the droplet slowdown velocity. As shown in Eq. 1, a small error in





the impact velocity leads to a large error in the erosion lifetime. The second central simplification is that, for the conclusions of this study, the exact droplet's shape *at impact* does not need to be predicted very accurately. This simplification is based on the assumption that an error in the droplet's shape during impact has a smaller influence on the erosion lifetime than an error in the impact velocity. However, at the same time, the prediction of the droplet's shape *prior to impact* needs to be accurate enough to minimize the error in the impact velocity. It is noteworthy that the droplet's shape at impact can be an input for a damage metric that is required to calculate an erosion lifetime. Thus, at first glance, a contradiction in the reasoning seems to occur. This aspect is discussed and resolved in Sect. 2.5.

Additional simplifying assumptions are made to model the problem in a Lagrangian one-particle setting. It is assumed that droplets will preserve a coherent shape during the entire approach towards the airfoil, i.e., not fracture, and thus can be represented as a single particle. This assumption neglects the potential effect of the non-homogeneous impact velocity of rain droplets during and after breakup. Based on the reference measurements from the literature that were presented before, it is also assumed that the cascade breakup does not occur.

Considering these requirements, the Droplet Ratio Deformation (DRD) model from Sor and García-Magariño (2015) was chosen. It was specifically developed to compute the trajectory of water droplets in the vicinity of approaching airfoils and stems from the same research group that has also published the measurements on droplet breakup discussed before. It has shown superior performance compared to other droplet models and is based on a one-way coupled Lagrangian approach. The original method uses three equations. One equation models the rain droplet's deformation from a sphere to the shape of an oblate spheroid. The other two equations model the movement of the droplet in a two-dimensional space. For the present study, the model was modified in such a way that the movement of the droplet can be considered one-dimensional only. It is important to note that the DRD model neither accounts for droplet breakup nor imposes any limit on the maximum deformation of a droplet. As a remedy, a heuristic modification is proposed in the following.

Two fundamental Equations of Motion (EOM) are at the model's core. They read

$$m\frac{d^2x}{dt^2} = F_{\text{drag}}, \tag{3}$$

$$\frac{3}{16}m\frac{d^2a}{dt^2} = F_\sigma + F_p. \tag{4}$$

Equation 3 represents the EOM along the droplet trajectory, whereas Eq. 4 is the EOM that represents the deformation of the droplet from a spheroid to an oblate spheroid. $m = 4/3\pi R_0^3 \rho_{\text{water}}$ is the mass of the droplet and $x$ is the position of the droplet along its trajectory. The possible candidates for the droplet trajectory will later be discussed in Sect. 2.3 together with Fig. 10. $a$ is the semi-major axis of an oblate spheroid, as shown in Fig. 8. $b$ is the semi-minor axis and can be calculated as $b = R_0^3/a^2$, where $R_0$ is the starting radius of a spherical rain droplet.



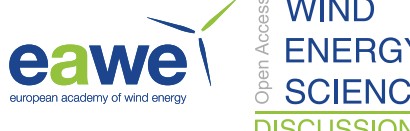

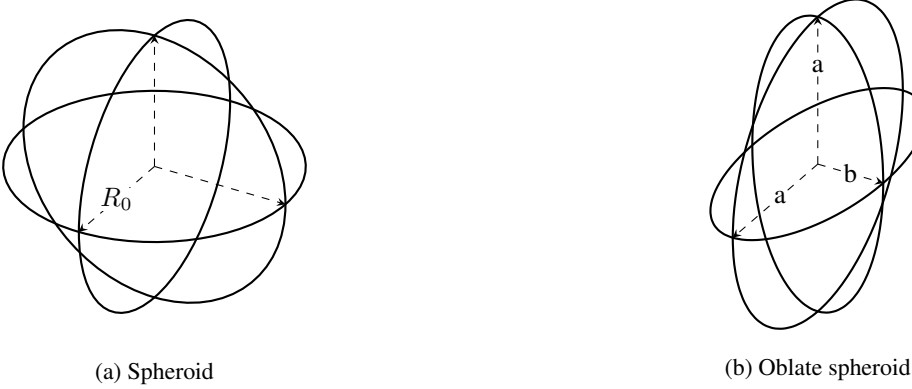

(a) Spheroid

(b) Oblate spheroid

**Figure 8.** Representation of the droplet shapes used in the model with the relevant geometrical parameters.

The drag force acting on the droplet is computed by using

$$F_{\text{drag}} = \frac{1}{2}\rho_{\text{air}}V_{\text{slip}}^2 C_D A_a. \tag{5}$$

$V_{\text{slip}}$ is the velocity difference between the air and the droplet; it reads

$$V_{\text{slip}} = V_{\text{air}} - \frac{dx}{dt}. \tag{6}$$

The calculation of the background velocity $V_{\text{air}}$ at a particular x is explained in Sect. 2.3. The droplet's instantaneous frontal area $A_a$ is calculated by simply taking $A_a = \pi a^2$. The drag coefficient is composed of a static and a dynamic component,

$$C_D = C_{\text{static}} + C_{\text{dynamic}}. \tag{7}$$

They read

$$C_{\text{static}} = C_{D_{\text{sphere}}}^{b/a} C_{D_{\text{disk}}}^{1-b/a}, \tag{8}$$

$$C_{\text{dynamic}} = k\frac{b}{V_{\text{slip}}^2}\frac{dV_{\text{slip}}}{dt}, \tag{9}$$

where $k$ is a calibration constant. The static component represents an interpolation between the drag coefficient of a sphere and a disk. In Eq. 4, two forces are acting against each other. The surface pressure term drives deformation, whereas the surface tension term counteracts deformation. The pressure term is calculated as

$$F_p = \frac{1}{2}\rho_{\text{air}}V_{\text{slip}}^2 C_p A_0. \tag{10}$$

$C_p$ is again a calibration constant. Also, note the constant surface area that is calculated with the initial droplet radius $R_0$. This choice is motivated in more detail in the original paper of the model. The surface tension force is written as

$$F_\sigma = -\frac{4}{3}\sigma_{\text{water}}\frac{dA}{da}, \tag{11}$$



where $\sigma_{\text{water}}$ is the surface tension of water and $\frac{dA}{da}$ is the derivative of the surface area of an oblate spheroid with respect to a. Following the approach of Sor and García-Magariño (2015, 2021); Schmehl (2004), the surface area of an oblate spheroid

reads

$$\frac{A}{A_0} = \frac{1}{2}\left(\frac{a}{R_0}\right)^2 + \frac{1}{2}\left(\frac{R_0}{a}\right)^4 \frac{\operatorname{arctanh}\epsilon}{\epsilon}, \tag{12}$$

with the derivative being

$$\frac{1}{A_0}\frac{dA}{d\bar{a}} = \bar{a} - \frac{2}{\bar{a}^5}\frac{\operatorname{arctanh}\epsilon}{\epsilon} + \frac{3}{2\bar{a}^5(\bar{a}^6-1)}\left(\bar{a}^6 - \frac{\operatorname{arctanh}\epsilon}{\epsilon}\right), \tag{13}$$

where

$$\epsilon = \sqrt{1 - \left(\frac{b}{a}\right)^2} = \sqrt{1 - \frac{1}{a^6}}, \tag{14}$$

and

$$\bar{a} = \frac{a}{R_0}. \tag{15}$$

Finally $\frac{dA}{da}$ is obtained by

$$\frac{dA}{da} = \frac{1}{R_0}\frac{dA}{d\bar{a}}. \tag{16}$$

$C_{D_{\text{sphere}}}$ has been calculated with the Schiller-Naumann relation as given in Sommerfeld et al. (2008),

$$C_{D_{\text{sphere}}} = \begin{cases} 27.6 & \text{Re} \leq 1, \\ \frac{24}{\text{Re}}\left(1 + 0.15\text{Re}^{0.687}\right) & 1 < \text{Re} < 1000, \\ 0.4383 & \text{Re} \geq 1000. \end{cases} \tag{17}$$

Note that the drag coefficient was clamped for $\text{Re} \leq 1$ and $\text{Re} \geq 1000$. The Reynolds number Re reads

$$\text{Re} = \frac{V_{\text{slip}}\rho_{air}2R_0}{\mu_{air}}. \tag{18}$$

In the original form, the model does not account for the influence of droplet breakup. The model permits the droplet to grow

without restriction. From the literature, such as Jackiw and Ashgriz (2021, 2022); Hsiang and Faeth (1995); Schmehl (2004), it is known that, depending on the Weber number, there exists a maximum diameter at which droplets start to break up. It usually lies in the range of 1.5 to 2 $a/R_0$. In this study, the assumption is made that when the droplets reach a specific maximum $a$, they will stop growing, and the value of $a$ will be fixed for the remainder of the simulation. In particular, the following formula is used

$$\frac{a_{\max}}{R_0} = \min\left(2.2, 3.4966\text{We}_\infty^{-0.1391}\right). \tag{19}$$





The formula was obtained by fitting an exponential curve to a set of reference data shown in Fig. 9. Further, the limit of 2.2 was chosen based on the data in Fig. 3 of García-Magariño et al. (2021). It is important to note that

$$\text{We}_\infty = \frac{\rho_\text{air} V_\infty^2 \phi_0}{\sigma_\text{water}}, \tag{20}$$

which is different to Eq. 2 since the free-stream velocity is used instead of the slip velocity. The motivation for this is the fact that for the limited sets of published data on $a_\text{max}/R_0$, the corresponding impact velocity is not always given. Additionally, $\text{We}_\text{impact}$ is not known a priori but rather a result of the simulation, therefore, necessitating an iterative approach for solving the set of equations. The assumption can be justified by realizing that $V_\text{impact} \approx V_\infty$ represents a conservative estimate. Since $a_\text{max}/R_0$ should be decreasing with increasing $\text{We}_\text{impact}$, assuming $\text{We}_\text{impact} \approx \text{We}_\infty$ will lead to a higher estimated Weber number. Thus, droplet slowdown will be underpredicted due to an underprediction in $a_\text{max}/R_0$. Section 2.4 shows that the

limiter introduced here deals with the droplet breakup satisfactorily.

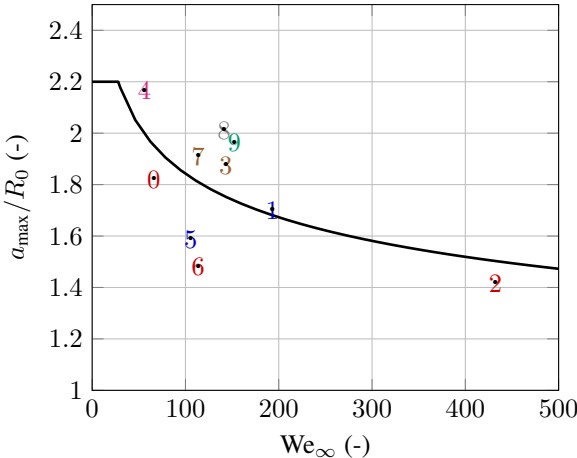

**Figure 9.** The limit of maximum droplet dimension $a_\text{max}/R_0$ as a function of free-stream Weber number; the sources of data points are given in Table 2; —— corresponds to Eq. 19.

**Table 2.** Sources of reference data for maximum droplet dimension $a/R_0$ limiter that is shown in Fig. 9.

| Symbol | Reference |
|---|---|
| 1 | Fig. 15 of Vargas et al. (2012) |
| 2 | Fig. 5.9 of García-Magariño (2016) |
| 3 | Fig. 8 and 9 of Vargas et al. (2012) |
| 4 | Table II of Feo et al. (2012) |
| 5 | Fig. 3.4 of García-Magariño (2016) |
| 6 | Fig. 3.9 of García-Magariño (2016) |
| 7 | Fig. A.3.4 of Sor (2017) |
| 8 | Fig. A.3.5 of Sor (2017) |
| 9 | Fig. A.3.6 of Sor (2017) |
| 0 | Table I of Feo et al. (2012) |

The resulting set of differential equations describing the droplet model is

$$\frac{dx}{dt} = V_x, \qquad\qquad \frac{dV_x}{dt} = \frac{F_\text{drag}}{m}, \tag{21}$$

$$\frac{da}{dt} = V_a, \qquad\qquad \frac{dV_a}{dt} = \frac{16}{3}\frac{F_\sigma + F_p}{m}. \tag{22}$$

The initial conditions for the droplet equations are set as

$$x_0 = 0, \qquad\qquad V_{x,0} = 0, \tag{23}$$

$$a_0 = R_0 + \text{eps}, \qquad\qquad V_{a,0} = 0, \tag{24}$$





'eps' is a very small number, e.g. 1E-12. This is necessary since Eq. 13 is not defined for $a = R_d$. An additional differential equation is needed to describe the movement of the blade. It reads

$$\frac{dx_{\text{blade}}}{dt} = V_{\text{blade}} = V_\infty = \text{const}, \tag{25}$$

with the initial conditions sufficiently far away from the droplet:

$$x_{\text{blade}} \gg R_c, \tag{26}$$

where a sufficiently far distance can, for example, be $20R_c$. The definition of $R_c$ is explained in Sect. 2.3. In this study, the differential equations were solved using a simple Runge-Kutta method. The simulation is stopped when the distance between the airfoil and the droplet falls below a certain threshold, i.e.

$$\Delta x = |x - x_{\text{blade}}| < \text{eps}, \tag{27}$$

where 'eps' is once again a small number. Table 3 summarizes the physical and calibrations constants used in the model. In the original method of Sor et al. (2016) $C_p$ was given as $C_p = 0.93$. However, in this study, it was found that setting $C_p = C_{D_{disk}}$ provided results that matched more closely the impact velocities of the validation cases in Fig. 18.

**Table 3.** Constants used in the model; physical properties at ambient temperature 288.15 Kelvin and ambient pressure of 101325 Pa.

| Constant | Value | Unit | Reference |
|---|---|---|---|
| k | 9 | (-) | (Sor et al., 2016) |
| $C_p$ | 1.17 | (-) | - |
| $C_{D_{disk}}$ | 1.17 | (-) | (Sor et al., 2016) |
| $\rho_{air}$ | 1.225 | kg m$^{-3}$ | - |
| $\mu_{air}$ | 1.7965E-5 | Pa s | - |
| $\rho_{water}$ | 999.1 | kg m$^{-3}$ | - |
| $\sigma_{water}$ | 0.07349 | N m$^{-1}$ | - |

**2.3 Calculation of the background velocity**

A necessary input to the model is the background velocity field $V_{\text{air}}$. The droplet traverses through this field while approaching the airfoil (see Eq. 6). It is dependent on the size and shape of the wind turbine's airfoil. This study treats the problem as one-dimensional. From this assumption, a range of possibilities for the implied trajectory of the droplet emerge. Figure 10 illustrates these possibilities. In the limit, there are two possible trajectories for small and large droplets, respectively. Very small droplets are expected to follow the streamline of the flow, while large droplets are expected to follow a ballistic trajectory. In practice,

the rain droplets will follow a trajectory that lies in the region between these two. To find a characteristic velocity field that





can be used for further study, two popular model turbine designs were chosen the NREL 5MW and the IEA 15MW (Jonkman et al., 2009; Gaertner et al., 2020).

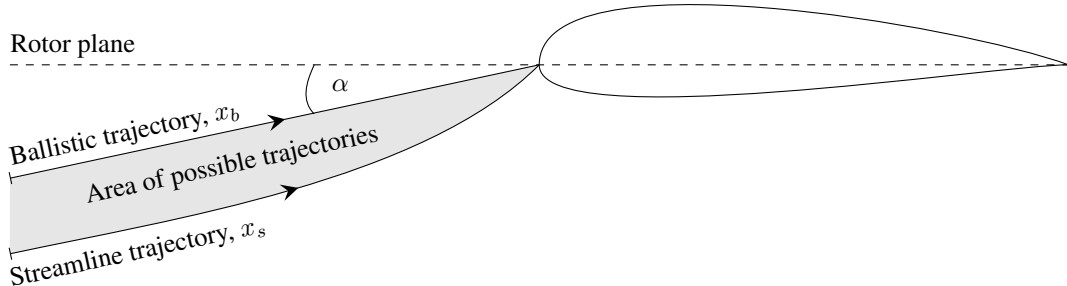

**Figure 10.** Ballistic and streamline trajectories of a droplet approaching an airfoil operated under an angle of attack $\alpha$; blade twist angle and pitch are set to zero.

The first step, taken here, towards obtaining $V_{\mathrm{air}}$ is to perform CFD calculations of the flow field surrounding the model turbines' airfoils using OpenFoam. The simulations were carried out by using the SIMPLE-Foam solver with the $k - \omega$ SST turbulence model. A free-stream velocity of 90 m s$^{-1}$ was chosen. A 2D mesh of around 100,000 cells has been used for the computations. In this application, a rather coarse computational grid is satisfactory since the area of interest is located ahead of the leading edge. In this area, the solutions are well-behaved and problematic areas with flow separation are located far downstream. Subsequently, the one-dimensional velocity field was extracted from the solution by using ParaView. Two fields were extracted, one for the ballistic trajectory and one for the streamline trajectory. The latter was obtained by seeding an upstream streamline from the leading edge in ParaView and subsequently extracting the velocity vector along this line.

Instead of directly using the extracted fields as a model input, they were parameterized, which allows to better compare the different airfoils by looking at the model parameters. As in Lopez-Gavilan et al. (2020), the underlying parametrization model is the potential flow solution of a cylinder representing the nose of the airfoil. The horizontal velocity component for the potential flow in the stagnation streamline reads

$$\frac{V_{\mathrm{air}}}{V_\infty} = 1 - \frac{1}{\left(1 - \frac{\Delta x}{R_{c,\alpha}}\right)^n}. \tag{28}$$

$R_{c,\alpha}$ is the radius of the cylinder and $\Delta x$ is the distance (from the droplet) to the cylinder. However, here $n$ and $R_{c,\alpha}$ are free parameters that are fitted to the extracted field from the CFD simulation. Therefore, $R_{c,\alpha}$ and $n$ should not be regarded as geometric but rather as *aerodynamic* parameters, i.e., $R_{c,\alpha}$ is the aerodynamic nose radius. It is also a function of the angle of attack. It is heuristically found that it is possible to collapse the 1-dimensional velocity field for different angles of attack, i.e., the solution is self-similar to an (arbitrary) scaling value. In this case, the self-similar variable is taken to be the distance from the leading edge at which the velocity has dropped to the 95 % value of the free-stream velocity ($x_{95\%}$). This self-similar





property is shown in Fig. 11. The left plots shows the velocity field against the dimensional distance to the leading edge. In the right plot the velocity field is collapsed by scaling with $x_{95\%}$.

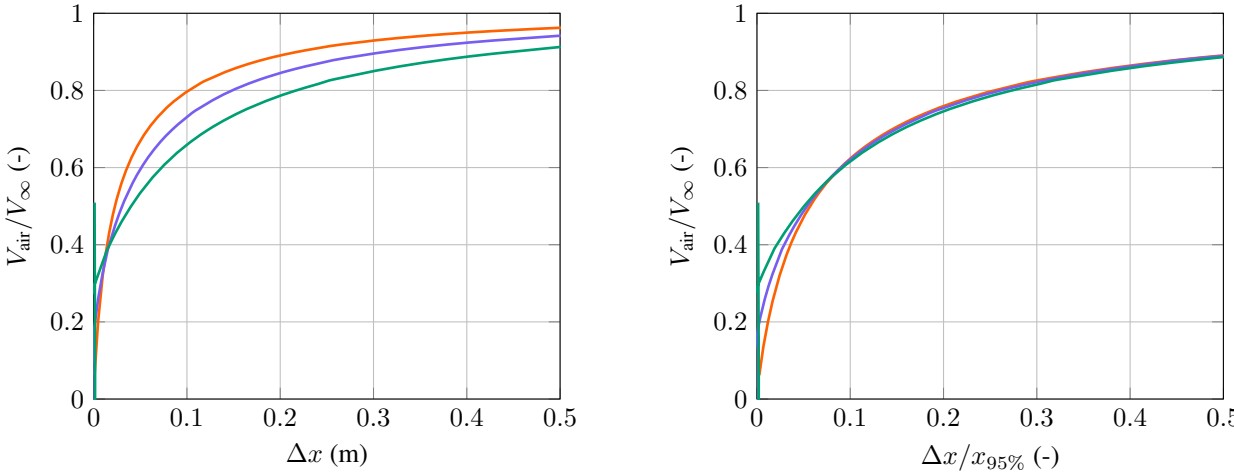

**Figure 11.** Non-dimensional velocity field along the stagnation streamline vs. the dimensional (left) and non-dimensional (right) distance to the leading edge; airfoil: FFA-W3-211; angle of attack: $0°$: ——, $7.5°$: ——, $15°$: ——.

The self-similarity allows to represent the velocity field at different angles of attack by scaling $R_{c,\alpha}$. The influence of the angle of attack variation on the self-similar parameters can be sufficiently represented by considering a second-order polynomial. Therefore,

$$x_{95\%,\alpha} = \left(C_1\alpha^2 + C_2\alpha + 1\right)x_{95\%,0}, \tag{29}$$

and thus also

$$R_{c,\alpha} = \left(C_1\alpha^2 + C_2\alpha + 1\right)R_{c,0}, \tag{30}$$

where $R_{c,0}$ is the aerodynamic nose radius at zero angle of attack. In Fig. 12 the FFA-W3-211 airfoil's variation of $x_{95\%,\alpha}$ is shown in conjunction with the polynomial fit. The first step in the parametrization process is to find $x_{95\%,\alpha}$ for every angle of attack. The zero-degree angle of attack field is then used to find the parameters $n$ and $R_{c,0}$. Last but not least, the parameters $C_1$ and $C_2$ are found by fitting the polynomial to $x_{95\%,\alpha}$. All best-fit parameters were found by using MATLAB's 'fmincon' function and a least squares minimization function.

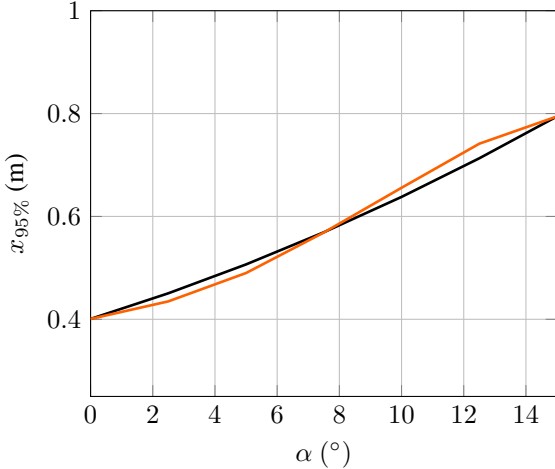

**Figure 12.** $x_{95\%}$ location of the FFA-W3-211 airfoil as a function of angle of attack; velocity field along the streamline trajectory; $x_{95\%}$ from CFD simulations: ——, quadratic fit of Eq. 30: ——.

The final parameters are given in Table 4. The values of $R_{c,\alpha}$ were made dimensionless with the airfoil chord $c$. The table shows a general trend when comparing thicker airfoils to thinner airfoils. Thicker airfoils have a higher aerodynamic nose radius and exponent. Therefore, the $x_{95\%}$ is also higher, meaning thicker airfoils influence droplets farther upstream. Two diverging behaviors can be noticed regarding the parameters for the angle of attack correction. For the flow that follows a ballistic trajectory, an increasing angle of attack leads to a decreasing $R_{c,\alpha}$, whereas for the flow along the stagnation streamline an

increasing $R_{c,\alpha}$ can be noticed. Therefore, in comparison to the zero-degree angle of attack, small droplets are expected to be influenced more, whereas large droplets are expected to be influenced less when the angle of attack is increased.





**Table 4.** Best-fit parameters of $V_{air}$ for the NREL 5MW and IEA 15MW turbine airfoils; subscript b and s stand for ballistic and streamline path, respectively; the coefficients $C_1$ and $C_2$ are dimensional; they are given for the angle of attack $\alpha$ in degrees, see Eq. 30; their units are given in the brackets of the column header.

| Airfoil | $R_{c,0}/c$ | $n$ | $C_{1,b}$ ($\circ^{-2}$) | $C_{2,b}$ ($\circ^{-1}$) | $C_{1,s}$ ($\circ^{-2}$) | $C_{2,s}$ ($\circ^{-1}$) |
|---|---|---|---|---|---|---|
| IEA 15MW Cylinder 2 | 0.3253 | 1.9542 | 3.17E-12 | -3.86E-11 | 1.84E-04 | -5.19E-04 |
| DU-99-W-405 | 0.1444 | 1.6546 | 5.29E-04 | -1.22E-02 | 6.94E-04 | 7.80E-03 |
| DU-99-W-350 | 0.0934 | 1.4508 | -1.25E-04 | -9.20E-03 | 8.09E-04 | 1.52E-02 |
| DU-97-W-300 | 0.0580 | 1.2708 | -5.17E-04 | -6.47E-03 | 1.16E-03 | 2.22E-02 |
| DU-91-W2-250 | 0.0414 | 1.1889 | -1.19E-03 | 2.22E-03 | 1.34E-03 | 3.20E-02 |
| DU-93-W-210 | 0.0297 | 1.1154 | -9.01E-04 | -3.71E-03 | 1.40E-03 | 4.47E-02 |
| NACA-64-618 | 0.0215 | 1.0494 | -8.86E-04 | -5.05E-03 | 1.34E-03 | 6.10E-02 |
| SNL-FFA-W3-500 | 0.2275 | 1.8662 | 4.15E-12 | -5.05E-11 | 5.10E-04 | -1.26E-03 |
| FFA-W3-360 | 0.1423 | 1.7035 | 6.13E-04 | -1.41E-02 | 8.36E-04 | 8.53E-03 |
| FFA-W3-330blend | 0.1114 | 1.5777 | 2.16E-04 | -1.37E-02 | 9.40E-04 | 1.90E-02 |
| FFA-W3-301 | 0.0804 | 1.4260 | -7.04E-04 | 2.85E-03 | 1.19E-03 | 2.16E-02 |
| FFA-W3-270blend | 0.0584 | 1.3084 | -6.49E-04 | -2.09E-03 | 1.58E-03 | 2.50E-02 |
| FFA-W3-241 | 0.0438 | 1.2227 | -1.19E-03 | 2.22E-03 | 1.85E-03 | 3.09E-02 |
| FFA-W3-211 | 0.0282 | 1.0974 | -9.01E-04 | -3.71E-03 | 1.61E-03 | 4.42E-02 |

Figures 13 and 14 show the dimensional aerodynamic nose radius $R_c$ and the exponent $n$ along the blade of the NREL 5MW and the IEA 15MW wind turbines. In line with the definition of both model turbines, the parameters were kept constant for different sections of the NREL turbine and were linearly interpolated across the different stations for the IEA turbine.

Figure 15 gives the angle of attack distributions along the blade that are used in Eq. 30. The influence on the angle of attack on the aerodynamic nose radius is less than 5 % for the ballistic trajectory and around 30 % for the streamline trajectory. Both $R_c$ and $n$ are larger for the IEA reference turbine than for the NREL design. This has three main reasons. The IEA turbine has a higher dimensionless aerodynamic nose radius $R_c/c$ for its airfoils. It also has a larger chord, and the angle of attacks are higher. Due to the similarity of the IEA 15MW turbine to current state-of-the-art off-shore turbines, it is argued that the

values of $R_c = 0.07$ m and $n = 1.1$, as they can be found at around $r/R = 0.9$, represent a good baseline for the remainder of this study. With these findings in mind, it is worth noting that with $R_c = 0.071$ m and $n = 1.2$, one obtains a very good fit of the reference velocity field of 'Case F' and 'G' (see Table 5). Hence, the parameter space of the reference data is close to the parameter space encountered in wind turbine erosion.





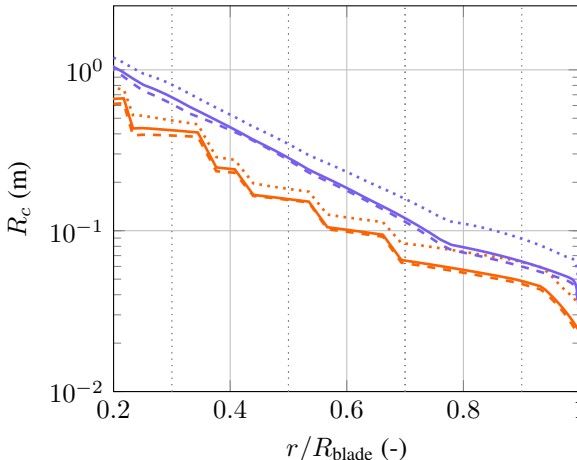
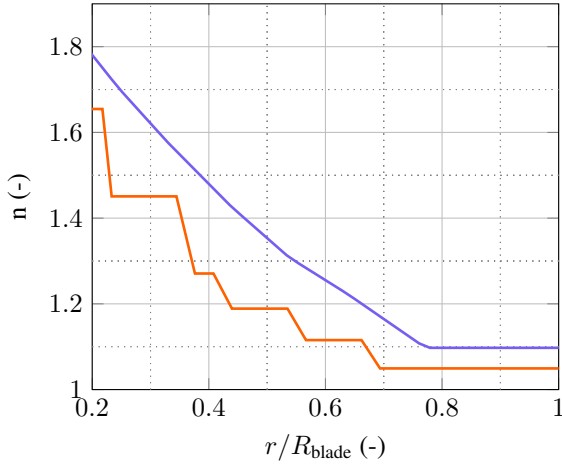

**Figure 13.** Dimensional aerodynamic nose radius $R_c$ along the dimensionless blade distance; IEA 15MW: No correction: ——, ballistic: – – –, streamline: ·······; NREL 5MW: No correction: ——, ballistic: – – –, streamline ·······.

**Figure 14.** Aerodynamic exponent $n$ along the dimensionless blade distance; IEA15 MW: ——; NREL5 MW: ——.

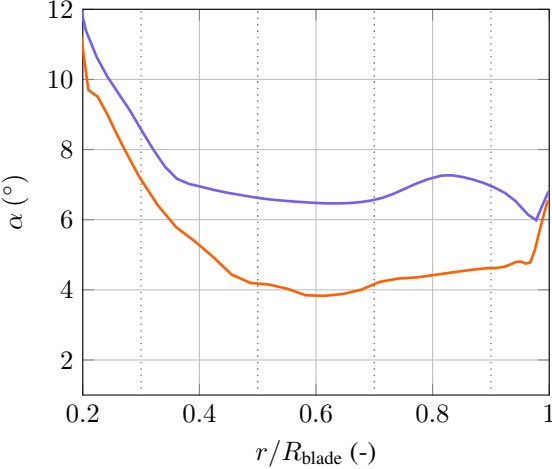

**Figure 15.** Elementwise angle of attack along $\alpha$ the dimensionless blade distance. The data were obtained from vortex method simulations in Barfknecht et al. (2022) with tip-speed ratio of 9 for the IEA 15MW and 7.55 for the NREL 5MW turbine; IEA 15MW: ——, NREL 5MW: ——.

## 2.4 Validation of the model

Two tests are performed to validate the model. First, the model is compared against well-known relations for the terminal falling conditions of water droplets. Secondly, a set of rotating-arm test-rig reference data is compiled from different sources.





Best (1950b) gives a relation for the terminal velocity of falling water droplets that reads

$$V_{\text{terminal}} = 9.32 e^{0.0405z} \left(1 - e^{-(0.565\phi_0)^{1.147}}\right), \tag{31}$$

where $z$ is the altitude in kilometers that was set to zero for this study. In this equation, $\phi_0$ must be given in millimeters. A

relation for the shape ($a/R_0$) of droplets at terminal conditions is given by Brandes et al. (2002). It reads

$$\frac{a}{R_0} = \left(0.9951 + 0.02510\phi_0 - 0.03644\phi_0^2 + 0.005030\phi_0^3 - 0.0002492\phi_0^4\right)^{-1/3}. \tag{32}$$

Note, in the original formulation, Eq. 32 was given as the ratio $a/b$. This has been converted here to $a/R_0$ by assuming the shape of an oblate spheroid. Also, $\phi_0$ must be given in millimeters for this equation. Figure 16 and 17 compare both formulas with the results from the model. An excellent agreement is achieved for the shape of the droplet. For up to 2.5 mm droplet

diameter, the terminal velocity is almost identical to the reference value. Afterwards, a slight deviation can be noticed.

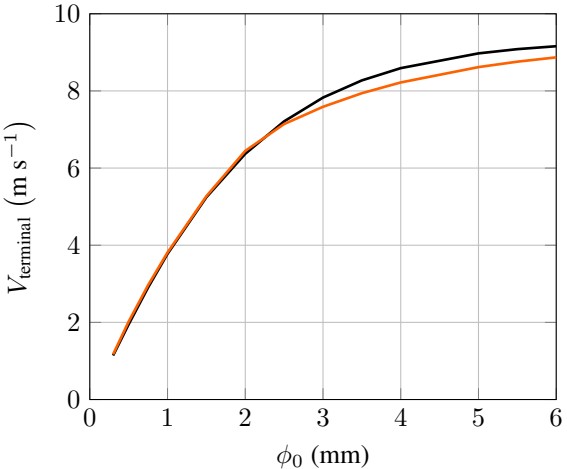
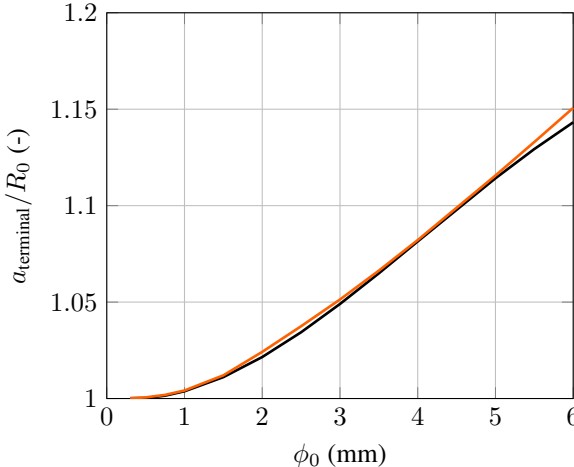

**Figure 16.** Terminal velocity for falling water droplets as a function of droplet diameter; simulation: ——; reference (Eq. 31): ——.

**Figure 17.** Terminal dimensionless semi-major axis for falling water droplets as a function of droplet diameter; simulation: ——; reference (Eq. 32): ——.

A set of reference data from the literature has been compiled for the second test. Unfortunately, the data quality differs based on whether they were directly available or had to be derived by, e.g., measuring distances on published images of high-speed photography. Table 5 summarizes the reference cases.

Figure 18 compares the model and measurements. It can be seen that there is a good agreement between the model and

the measurements. The model overpredicts the slowdown for Case F. Interestingly, the slight discrepancy starts already at a distance of about 0.05 m from the leading edge, a region where the other cases show excellent agreement. Cases D and E suffer from a slight underprediction of the slowdown close to the leading edge. Arguably, Case D overpredicts the slowdown. Data extraction of Case H was challenging and had to be done manually from a small series of published photographs. Therefore, the data can only be considered fair. Nevertheless, the simulation and measurements still agree reasonably well.



**Table 5.** Summary of rotating-arm test-rig reference data used in the validation of the proposed model.

| Name | $\phi_0$ (mm) | $V_\infty$ $\left(\mathrm{m\,s^{-1}}\right)$ | c (m) | Data | Source |
|---|---|---|---|---|---|
| Case A | 0.490 | 50 | 0.47 | extracted from graph | Fig. 25 from Vargas and Feo (2011) |
| Case B | 0.490 | 60 | 0.47 | extracted from graph | Fig. 25 from Vargas and Feo (2011) |
| Case C | 0.490 | 70 | 0.47 | extracted from graph | Fig. 25 from Vargas and Feo (2011) |
| Case D | 0.490 | 80 | 0.47 | extracted from graph | Fig. 25 from Vargas and Feo (2011) |
| Case E | 0.490 | 90 | 0.47 | extracted from graph | Fig. 25 from Vargas and Feo (2011) |
| Case F | 1.062 | 90 | 0.71 | extracted from graphs | Fig. 5, 10 from Vargas et al. (2012) |
| Case G | 1.431 | 90 | 0.71 | derived from multiple graphs | Fig. 10, 14, 15 from Vargas et al. (2012) |
| Case H | 3.201 | 90 | 0.69 | from measuring features in images | Fig. 5.9 from García-Magariño (2016) |

To summarize, the model agreed well with reference data for both validation cases. Recall, even slight differences in the impact speed will lead to very different lifetime predictions due to the large exponent in Eq. 1. Nevertheless, with the available data and the simple reduced-order Lagrangian model in mind, the validation results are considered adequate for lifetime predictions.



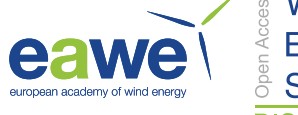

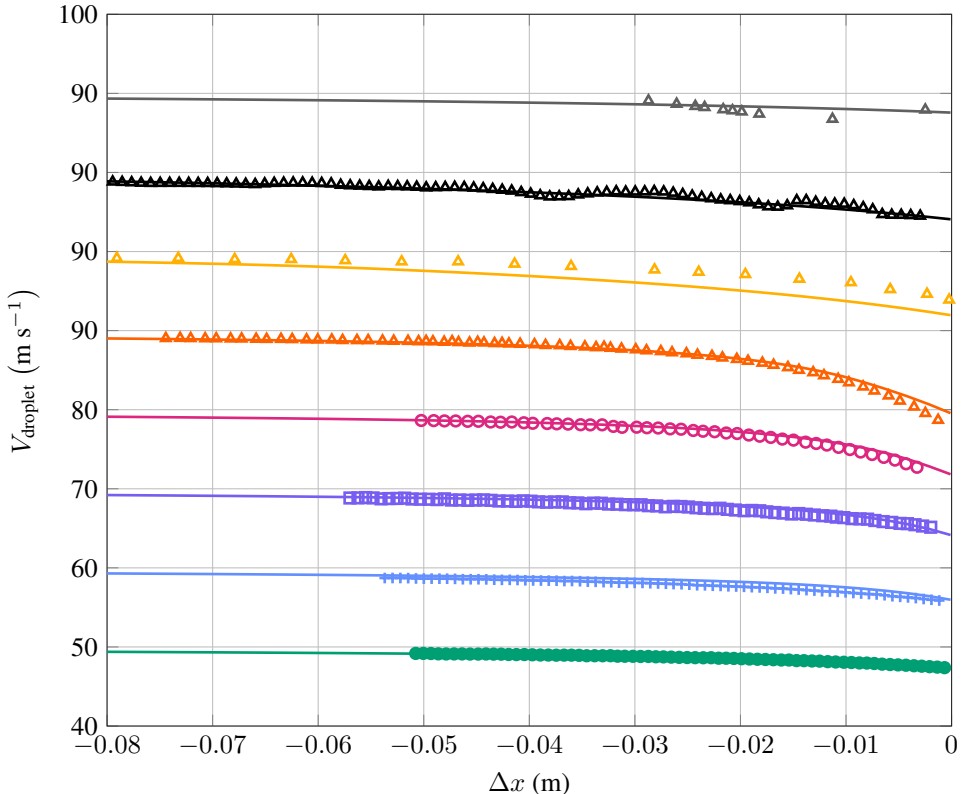

**Figure 18.** Validation of the trajectory model with reference data summarized in Table 5; markers indicate reference data and solid lines results of the model; note, the $y$-axis contains repeated ticks for better visualization of cases with equal free-stream velocity; Case A: ——, Case B: ——, Case C: ——, Case D: ——, Case E: ——, Case F: ——, Case G: ——, Case H: ——.

## 2.5 Damage model and relevant rain droplet diameters

A damage model is required to evaluate the magnitude of the slowdown effect on the lifetime of a blade. The damage model proposed in this paper is described in the following. Additionally, the equations developed here are also used to compute the relevant rain droplet diameter range for the present study.

Several damage metrics have been proposed to calculate an erosion lifetime: the water hammer pressure metric, which is often used in conjunction with the Springer model (Hoksbergen et al., 2022), impingement (Bech et al., 2022), kinetic energy 325 (Bech et al., 2018) or the material's strain (Verma et al., 2020). Arguably, the two most common models are currently the Springer model and the impingement metric. This study uses the impingement metric to calculate an erosion lifetime. The choice is motivated in the following.

The Springer model (as described in Hoksbergen et al. (2022)) gives an equation for the erosion lifetime by considering the number of allowable repeated impacts on one location $N_i^*$. The model is derived by computing an impact force $F = $ 330 $p_{wh} A_{projected}$, where $p_{wh}$ is the (modified) water hammer pressure and $A_{projected}$ is the projected area of an impacting droplet.





The assumption is made that the water hammer pressure is constant for the entire projected area of the droplet. Subsequently, a stress field within the target is computed using an analytical equation of the form $\sigma(F, r, ...)$, where $r$ is the distance to the impact location. Further, $\sigma \propto F$. After some steps, $N_i^*$ is obtained. The entire derivation for the (uncoated) Springer model is given in Springer and Baxi (1972). One of the problematic assumptions within the Springer model is the calculation of the impact force. If, for example, a droplet is infinitely stretched, that is $A_{\text{projected}} \to \infty$, then $\sigma \to \infty$ and therefore $N_i^* \to 0$. Alternatively, a droplet that is squeezed, i.e. $A_{\text{projected}} \to 0$, will have a lifetime of $N_i^* \to \infty$. Both results seem unphysical and thus question the validity of the Springer model. Since the rain droplets deform significantly and, therefore, grow in the projected area, the Springer model does not seem to be an adequate choice for the present study.

Impingement is a damage metric representing the total water column the blade intercepts until coating failure. Since impingement only considers the amount of water, it is, at least conceptually, agnostic to the impacting droplet's shape; a property that seems advantageous considering the complex shape of droplets during impact. Due to this property and its recent gain in popularity, as shown in Bech et al. (2022); Visbech et al. (2023); Badger et al. (2022), it was chosen as the damage metric for this study.

The general formula for the accumulated impingement $H$ during operation is

$$H = W V_{\text{collection}} t. \tag{33}$$

$W$ is the accumulated water column in meters per meter of swept air, $t$ is the time, and $V_{\text{collection}}$ is the speed at which water is collected. Here it is assumed that $V_{\text{collection}} = V_\infty$.

The *impingement until end of incubation*, dubbed *allowed impingement*, is also required. $H_{\text{allowed}}$ reads

$$H_{\text{allowed}} = \frac{\alpha}{V_{\text{impact}}^\beta}. \tag{34}$$

The equation has the form of Eq. 1. The parameters $\alpha$ and $\beta$ were found using the measurements of Bech et al. (2022). They performed measurements in a rotating-arm erosion test-rig, where they recorded $H_{\text{allowed}}$ of a generic blade coating with respect to $V_{\text{impact}}$. Tests with four distinct droplet sizes ranging from 0.76 to 3.5 mm were performed. The measurements are shown in Fig. 19. Their raw data were used in this study to fit a function through the data points, leading to the best-fit parameters of $\alpha = 3.4860E20$ and $\beta = 9.5774$. The figure shows that the measurements collapse well. It should be noted that the authors of the study argue that the data show some drop-size dependency with coefficients of $\beta$ being in the range of 7.2-10.5. This range is found when a best-fit function is created for every droplet size individually. Nevertheless, the assumption made here is that this dependency can be neglected for the conclusions drawn in this study.

Since, as indicated earlier, there is a wide spread of reported values for $\beta$ in the literature, two other exponents were considered to ensure the robustness of the drawn conclusions with respect to $\beta$. The other two exponents that were chosen are 5.7 and 7. The exponent of 5.7 originates from the Springer model (Hoksbergen et al., 2022). Even though Springer does not measure impingement, but rather impacts (per surface area), it is still considered to be worth showing. The exponent 7 represents an arbitrary value between 5.7 and 9.5774.





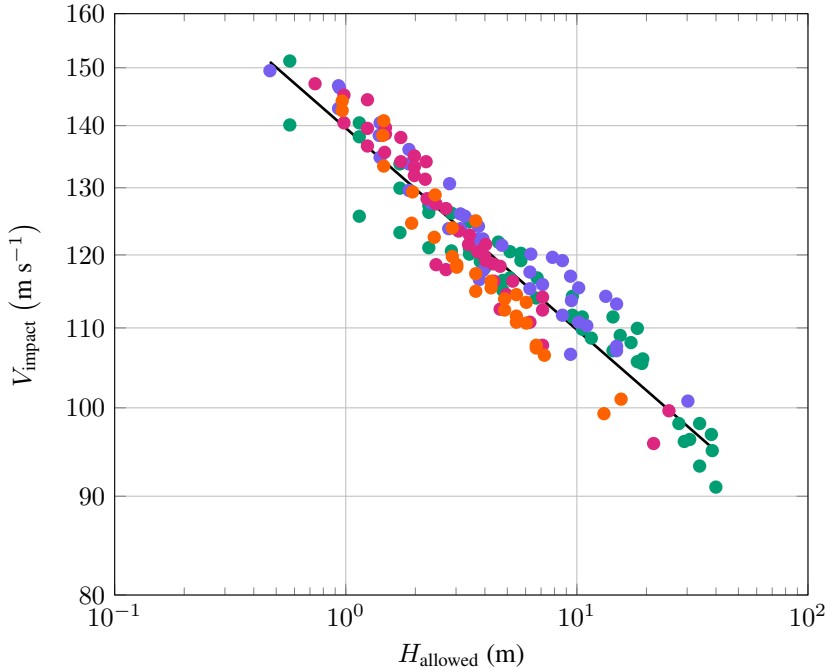

**Figure 19.** Rotating-arm erosion test-rig results by Bech et al. (2022) that relate impact velocity to impingement; droplet diameters are 0.76 mm: ●, 1.90 mm: ●, 2.38 mm: ●, 3.50 mm: ●; fit of all diameters: ——.

Equation 33 and 34 can be used in a Palmgren-Miner damage (PMD) rule, giving

$$\text{PMD} = \int\limits_{0}^{\infty}\int\limits_{0}^{\infty} \frac{H_{I,\phi}}{H_{\text{allowed}}(V_{\text{impact}}(\phi))}\,d\phi dI. \tag{35}$$

$H_{I,\phi}$ is the impingement as a function of the rain intensity $I$ and $\phi_0$. It is defined as

$$H_{I,\phi} = W_{I,\phi}V_{\text{collection}}t_I, \tag{36}$$

and is analogous to Eq. 33, but with $W_{I,\phi}$, which is again a function of $I$ and $\phi_0$. Here it should be noted that this equation is only valid for the 12 and 6 o'clock positions of the wind turbine blade. However, for the conclusions that will be drawn here, this simplification is deemed acceptable. $W_{I,\phi}$ is given by

$$W_{I,\phi} = \frac{f_{\phi,\text{plane}}I}{V_{\text{terminal}}}, \tag{37}$$

where $f_{\phi,\text{plane}}$ is a distribution that describes the amount of water associated with every droplet diameter that passes through an imaginary plane in the air. One popular model that can be used to obtain $f_{\phi,\text{plane}}$ is the Best model (Best, 1950a). It gives a probability density function (pdf) of the water mass associated with every droplet diameter in a control volume in air and is

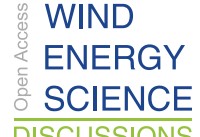

given as

$$f_{\phi,\text{air}} = 2.25 \left( \frac{1}{1.3 I^{0.232}} \right)^{2.25} \phi_0^{2.25-1} e^{-\left( \frac{\phi_0}{1.3 I^{0.232}} \right)^{2.25}},$$

(38)

where here, notice the units, $I$ is the rain intensity in millimeters per hour and $\phi_0$ is the droplet diameter in millimeters! $f_{\phi,\text{air}}$ can be converted into $f_{\phi,\text{plane}}$ by using

$$f_{\phi,\text{plane}} = \frac{f_{\phi,\text{air}} V_{\text{terminal}}}{\int_0^\infty f_{\phi,\text{air}} V_{\text{terminal}} d\phi}.$$

(39)

Note that if $f_{\phi,\text{plane}}$ is supposed to be obtained for droplet diameters in meters, then the integral in the denominator should be computed with $\phi$ in meters. $f_{\phi,\text{plane}}$ is plotted for five different rain intensities in Fig. 20. One can see that the water volume of lighter rain events is mainly composed of droplets with smaller diameters in the order of 0.5 to 1 mm. With increasing rain intensity, the amount of water contained in larger droplets is increasing.

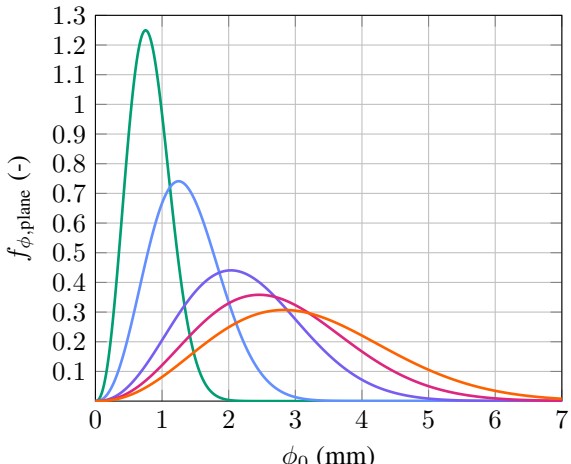

**Figure 20.** Best's distribution over a plane as a function of droplet diameter in mm for five different rain intensities; rain intensities 0.1 mm hr$^{-1}$: ——, 1 mm hr$^{-1}$: ——, 10 mm hr$^{-1}$: ——, 25 mm hr$^{-1}$: ——, 50 mm hr$^{-1}$: ——.

The collection time $t_I$ over one year of operation is given by

$$t_I = T_{\text{year, spinning}} p_{\text{rain}} f_I,$$

(40)

where $T_{\text{year, spinning}}$ is the number of seconds in a year that the turbine spins and $p_{\text{rain}}$ is the probability of rain at a particular site. It should be noted that in the results section of this study the damage is presented in its non-dimensional form. $p_{\text{rain}}$ and $T_{\text{year, spinning}}$ cancel during non-dimensionalization since they are both constant. $f_I$ is the probability density function for the various rain intensities. To find $f_I$, in this study, we consider the coastal site De Kooy located in The Netherlands at coordinates (52.924, 4.780). Hourly precipitation data from a 10-year window from 2011 to 2020 were used (KNMI -





Koninklijk Nederlands Meteorologisch Instituut, 2020). The probability density function was determined by using the same
approach as in Shankar Verma et al. (2021) where a lognormal distribution that reads

$$f_I = \frac{1}{I\sigma\sqrt{2\pi}} e^{-\frac{(\ln I - \mu)^2}{2\sigma^2}} \tag{41}$$

was fitted using Matlab's 'lognfit' function to the measured precipitation data of the site. $\mu$ is the mean and $\sigma$ is the standard
deviation. Note here the different meanings of the symbols in comparison to before. For a rain intensity given in mm hr$^{-1}$, the

395 coefficients read $\mu = -0.1987$ and $\sigma = 0.9693$, whereas when $I$ is given in m s$^{-1}$, the coefficients become $\mu = -15.29$ and
$\sigma = 0.9693$.

By combining the previous equations one obtains the universal Palmgren-Miner damage for an element along the blade
reading

$$\text{PMD} = T_{\text{year, spinning}} p_{\text{rain}} V_{\text{collection}} \int_0^\infty I f_I \int_0^\infty \frac{f_{\phi,\text{plane}}/V_{\text{terminal}}}{H_{\text{allowed}}(V_{\text{impact}}(\phi))} d\phi dI. \tag{42}$$

The formula written in its cumulative form reads

$$\text{PMD}_{\text{cumulative}} = T_{\text{year, spinning}} p_{\text{rain}} V_{\text{collection}} \int_0^I I' f_{I'} \int_0^\infty \frac{f_{\phi,\text{plane}}/V_{\text{terminal}}}{H_{\text{allowed}}(V_{\text{impact}}(\phi))} d\phi dI'. \tag{43}$$

A special version can be derived that gives the damage associated per meter of impingement at a particular rain intensity.
Non-dimensionalizing $H_{I,\phi}$ yields

$$\overline{H_{I,\phi}} = \frac{H_{I,\phi}}{\int_0^\infty H_{I,\phi} d\phi} = \frac{f_{\phi,\text{plane}}/V_{\text{terminal}}}{\int_0^\infty f_{\phi,\text{plane}}/V_{\text{terminal}} d\phi} = f_{\phi,\text{air}}. \tag{44}$$

Since

$$\int_0^\infty \overline{H_{I,\phi}} d\phi = 1, \tag{45}$$

the damage associated per meter of impingement becomes

$$\text{PMD}_{1m,I} = \int_0^\infty \frac{f_{\phi,\text{air}}}{H_{\text{allowed}}(V_{\text{impact}}(\phi))} d\phi. \tag{46}$$

Equation 42 can be rewritten with Eq. 46 which yields

$$\text{PMD} = \int_0^\infty \text{PMD}_{1m,I} \int_0^\infty H_{I,\phi} d\phi dI, \tag{47}$$

where the first part is the damage per meter of impingement at a particular rain intensity. She second part is the meter of
impingement at a particular rain intensity.



The distribution of water mass that is associated with every droplet diameter at a particular site can be found by combining the functions of $f_{\phi,\text{plane}}$ and $f_I$. The result reads

$$f_{\phi,\text{site}} = \frac{\int_0^\infty I f_{\phi,\text{plane}} f_I dI}{\int_0^\infty \int_0^\infty I f_{\phi,\text{plane}} f_I dI d\phi}, \tag{48}$$

with the corresponding cumulative density function of

$$F_{\phi,\text{site}} = \int_0^\phi f_{\phi',\text{site}} d\phi'. \tag{49}$$

Both of these functions are plotted in Figure 21. It can be seen that the droplets in the range of 0 to 4 mm contain around 99 % of the total water content. This range needs to be studied for the slowdown effect. The droplets in the range of 0.5 to 3.0 mm account for about 92.5 % of water. $F_{\phi,\text{site},50\%}$ is found at a diameter of 1.54 mm.

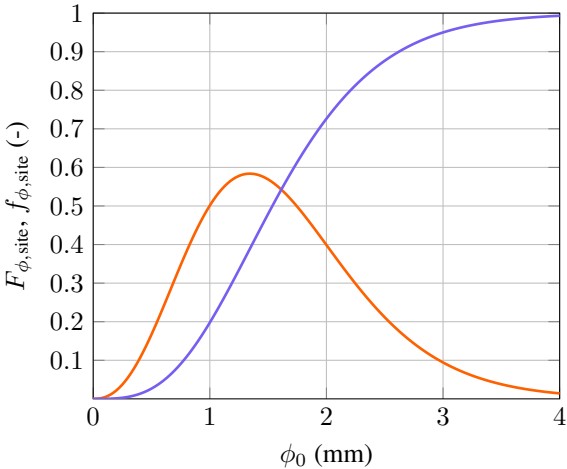

**Figure 21.** Distribution and cumulative function of the total rain column associated with every droplet diameter; distribution $f_{\phi,\text{site}}$: ——, cumulative $F_{\phi,\text{site}}$: ——.

## 3   Results

In this part, the slowdown and deformation model from the previous section is applied. First, in Sect. 3.1 and 3.2, the sensitivities of the droplet diameter and the aerodynamic nose radius on the slowdown and deformation are investigated. Subsequently, in Sect. 3.3, the model's influence on the erosion damage associated with rain intensities is determined. In Sect. 3.4, the distribution of the slowdown velocity along two model turbines is discussed. Finally, these velocities are used to determine an updated damage distribution along the blades of the model turbine.



## 3.1 The influence of droplet diameter and shape on the impact velocity

This section discusses the sensitivity of the droplet slowdown with respect to its diameter. Two types of droplets are considered, spherical and oblate droplets. The results of the spherical droplets serve as a conservative bound to the problem and represent the minimum slowdown. As per Fig. 21, droplets in the range of 0.1 to 4 mm were considered for free-stream velocities ranging from 50 to 90 m s$^{-1}$.

Figure 22a shows the dimensional impact velocity of spherical droplets. A significant slowdown of the droplets can be observed for droplets under 0.5 mm diameter. Larger droplets show a more gradual slowdown. The origin of this behavior can be found in the ratio of surface area to mass, which is much larger for smaller droplets, thus making them more affected by the drag force. Non-dimensionalizing the impact velocity reveals that the impact velocity for spherical droplets is self-similar, as shown in Fig. 22b. The curves collapse onto each other, indicating common slowdown factors for every drop diameter, irrespective of the free-stream velocity. Therefore, the absolute slowdown for faster droplets will be larger than their slower peers.

The results for the deformed droplets, as shown in Fig. 22c, reveal additional effects. First, it can be observed that the impact velocities are noticeably lower. For example, droplets of 1 mm diameter and 90 m s$^{-1}$ free-stream velocity are slowed down by around 2.5 m s$^{-1}$ when kept spherical, whereas deformation leads to a slowdown of about 10 m s$^{-1}$. The reason for this is that the larger surface area due to the deformation leads to higher drag forces, increasing the slowdown for oblate droplets. The impact velocity graphs of the spherical droplets have a concave shape. In the graphs of the oblate droplets, a saddle point appears in the region of 0.5 mm diameter. The prominence of this saddle point increases with increasing free-stream velocities. From 70 m s$^{-1}$ the impact velocity is not monotonically increasing but shows a slight dip at the saddle point. It is, therefore, possible that a larger droplet has a lower impact velocity. The location of the saddle point coincides approximately with the maximum deformation of the droplet, as shown in Fig. 22e. In this figure, the deformation is shown to rise to a maximum, after which it begins to decline. The maximum corresponds to the diameter at which the limiter of Eq. 19 starts to restrict the growth of the droplets. However, the limiter is not the reason for the occurrence of the saddle points. This can be shown by simulations without limiter where the prominence and extent of the saddle point grows. Therefore, the saddle point must be a consequence of the non-linear coupling of the momentum and deformation equation and cannot be attributed to the limiter. It would be interesting to know whether this saddle point can also be observed in experiments. The non-dimensional impact velocity of oblate droplets is self-similar outside the region of the saddle points. In the region of the saddle point, the non-dimensional impact velocities are lower for higher free-stream velocities, indicating that an extra slowdown is obtained greater than the common scaling factors of the self-similar solution. It is also evident that with increasing free-stream velocities, the overlap of the curves becomes larger, meaning that, for example, the solutions of 80 m s$^{-1}$ and 90 m s$^{-1}$ are more self-similar than the ones of 50 m s$^{-1}$ and 90 m s$^{-1}$. It can be summarized that oblate droplets slow down more than their spherical peers and that the slowdown effect is sensitive with respect to the droplet diameter.





(a) Spherical dropletss; dimensional velocity.

(b) Spherical droplets; non-dimensional velocity.

(c) Oblate droplets; dimensional velocity.

(d) Oblate droplets; non-dimensional velocity.

(e) Oblate droplets; droplet semi major axis.

**Figure 22.** Impact velocity for different droplet diameters and free-stream velocities; aerodynamic nose radius $R_c = 0.07$ m, exponent $n = 1.1$; $V_\infty$ of 50 m s$^{-1}$: ——, 60 m s$^{-1}$: ——, 70 m s$^{-1}$: ——, 80 m s$^{-1}$: ——, 90 m s$^{-1}$: ——.





## 3.2 The influence of the aerodynamic nose radius on the impact velocity

The influence of the aerodynamic nose radius on the impact speed was investigated for a combination of spherical and oblate droplets of 0.5 and 2 mm diameter. Figure 23 shows that 0.5 mm droplets are much more sensitive to a change in $R_c$ than the larger droplets of 2.0 mm. For example, spherical droplets of 0.5 mm diameter have their normalized impact velocity reduced by about 0.1 when $R_c$ is increased from 0.1 to 0.2 m. The impact velocity of the 2.0 mm droplets decreases in the same range by only about 0.01. In general, the curves of the spherical droplet are closely overlapping, indicating again a self-similarity.

Oblate droplets show much greater sensitivity towards $R_c$, as seen when comparing Fig. 23c and 23d. Over the entire range of the investigated nose radii, the velocities of the spherical droplets of 2.0 mm decrease by about 0.05, whereas a decrease of approximately 0.25 to 0.3 can be observed for the oblate droplets, i.e. a factor of five larger. No self-similarity can be observed for oblate droplets of 0.5 mm diameter as shown in Fig. 23b. It is worth nothing that the differences with respect to the free-stream velocity among 0.5 mm droplets increase with increasing $R_c$. The consequence is that, in rotating-arm test

rigs with small-scale airfoils, the dependency on the free-stream velocity is most likely not captured. The free-stream velocity dependency originates from the non-linearity, as discussed in the previous section together with Fig. 22d. For oblate droplets of 2.0 mm, the curves again overlap closely, as was also the case in Fig. 22d. Therefore, droplets in the saddle point region are especially sensitive to a change in the nose radius. This property is interesting since it means that, especially for faster tip speeds, a higher $R_c$ gives extra slowdown and thus reduces blade damage. Therefore, from a mitigation perspective, it appears

to be attractive to utilize *aerodynamically thicker* airfoils; see Table 4. To summarize, the slowdown effect for oblate droplets is highly sensitive to the aerodynamic nose radius. This sensitivity provides an interesting opportunity as an erosion mitigation strategy.





**Figure 23.** Non-dimensional droplet impact velocity for different aerodynamic nose radii $R_c$ and free-stream velocities; exponent, $n = 1.1$; $V_\infty$ of 50 m s$^{-1}$: ———, 60 m s$^{-1}$: ———, 70 m s$^{-1}$: ———, 80 m s$^{-1}$: ———, 90 m s$^{-1}$: ———.

### 3.3 Erosion damage associated with rain intensities and its distribution

This section evaluates the effect of the droplet slowdown on the PMD equations from Sect. 2.5. First, Eq. 46 is considered,

which gives the damage associated with 1 m of impingement at a particular rain intensity. The average droplet impact speed must vary with rain intensity since every rain intensity has a distinct drop-size distribution. As a result, equal amounts of impingement originating from different rain intensities lead to varying degrees of damage. Without the slowdown effect, the impact speed of all droplets, irrespective of their diameter, is equal, and there will be no distinction in damage across the rain intensities. Note that the terminal velocity of a droplet and its dependency on the diameter is neglected here. The PMD damage





of Eq. 46 was brought into a non-dimensional form with

$$\overline{\mathrm{PMD}_{1m,I}(I,\beta)} = \frac{\mathrm{PMD}_{1m,I}(I,\beta)}{(\mathrm{PMD}_{1m,I}(I,\beta))_{\text{no slowdown}}}. \tag{50}$$

Three distinct damage exponents $\beta$ from Sect. 2.5 were considered to establish the robustness of the results with respect to the damage metric. The results are shown in Fig. 24. Droplets without slowdown are non-dimensionalized with themselves and, thus, show a damage of unity in the entire plot. The damage for spherical and oblate droplets varies with rain intensity. At low rain intensity, most water mass is contained in the smaller droplets, which experience a significant slowdown, as shown before in Fig. 22a and 22c. Therefore, low rain-intensity rain shows a large reduction in its damage. As the rain intensity increases, so does the fraction of large droplets within the rain. The large droplets experience considerably less slowdown and, thus, are much more damaging.

Even though the exponents span a wide range, the spherical and oblate droplets' curves remain close together with respect to themselves. The difference in damage between the highest and the lowest exponent is fairly constant for both droplets across the entire range of rain intensities. This difference is approximately 0.1 and 0.175 for spherical and oblate droplets, respectively. Spherical droplets, especially for smaller rain intensities, already show so much damage reduction that the slowdown effect cannot be neglected. The difference in damage between spherical and oblate droplets is even more significant than between spherical droplets with and without slowdown. Thus it is not sufficient to assume that droplets are spherical, but the deformation needs to be taken into account as well. Figure 24 also shows $f_I$, which is the pdf of the rain intensities. Around 80 % of all precipitation events are of the magnitude 2 mm hr$^{-1}$ and lower. In this range, the slowdown also has the highest effect.

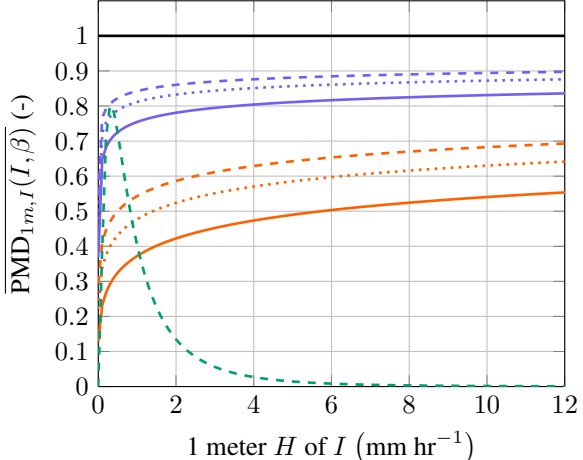

**Figure 24.** Normalized erosion damage for 1 m of rain impingement at different rain intensities; default parameters of $V_\infty = 90$ m s$^{-1}$, $R_c = 0.07$ m, $n = 1.1$; no slowdown: —; spherical droplets: $\beta = 5.7$: - - -, $\beta = 7$: ······, $\beta = 9.58$: —; oblate droplets: $\beta = 5.7$: - - -, $\beta = 7$: ······, $\beta = 9.58$: —; normalization reference is with respect to no-slowdown droplets.





Equation 46 can be combined with $f_I$ to obtain the total expected erosion damage at a particular wind turbine site. Doing so yields Eq. 47 and its cumulative form Eq. 43. In its non-dimensional form, the cumulative site damage reads

$$\overline{\text{PMD}_{\text{cumulative}}(\beta)} = \frac{\text{PMD}_{\text{cumulative}}(\beta)}{(\text{PMD}(\beta))_{\text{no slowdown}}}. \tag{51}$$

It is plotted in Fig. 25a for all three droplet types and damage exponents. Since the damage is written in its cumulative form, the damage of droplets without slowdown reaches unity for $I \to \infty$. The plot shows that for a turbine located at the De Kooy weather station, the inclusion of the droplet slowdown leads to predicted damages of 0.77 to 0.85 for spherical and 0.41 to 0.57 for oblate droplets. Or expressed in the reciprocal (i.e., the lifetime), droplet slowdown with oblate droplet leads to about two times longer lifetimes depending on the damage exponent. Figure 25a also shows which rain intensities contribute the most to 510 erosion damage. E.g. for droplets without slowdown, all rain events between 0 and 2 mm hr$^{-1}$ contribute to about 55 % of the total erosion damage. From this, the question arises whether the slowdown also influences which rain intensities contribute the most towards erosion damage. To study this, a different non-dimensionalization is used

$$\overline{\text{PMD}_{\text{cumulative, relative}}(\beta)} = \frac{\text{PMD}_{\text{cumulative}}(\beta)}{\text{PMD}(\beta)}. \tag{52}$$

Here every case is non-dimensionalized with itself so that the erosion damage for $I \to \infty$ is always unity. The results are shown 515 in Fig. 25b. For oblate droplets, the 55 % mark of relative damage is shifted to around 2.3 mm hr$^{-1}$ compared to 2 mm hr$^{-1}$ for the case without slowdown. This shows that the slowdown effect not only significantly reduces the predicted erosion damage but also slightly shifts the "production" of erosion damage to higher rain intensities.

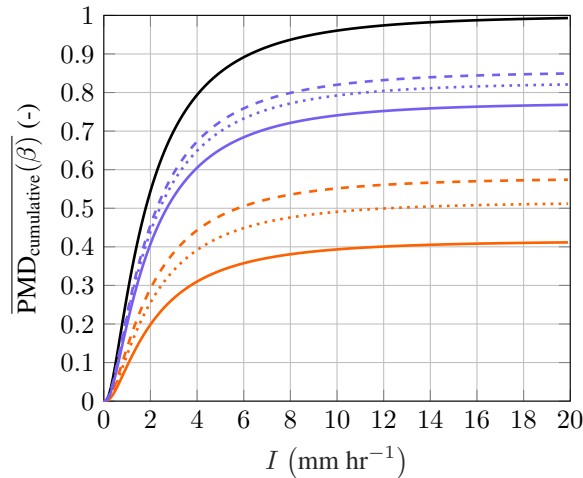
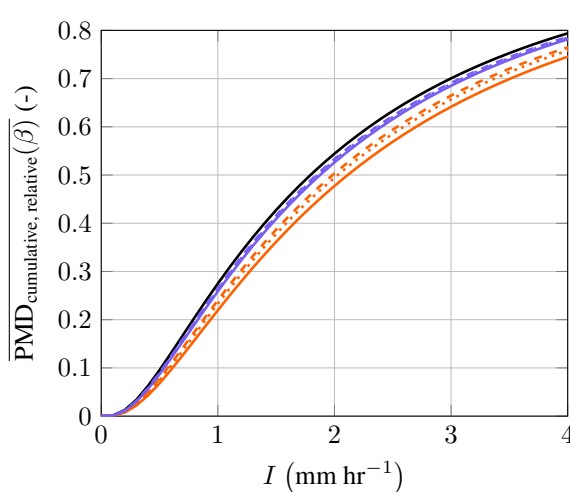

(a) Normalization with respect to no slowdown droplets. See Eq. 51.   (b) Normalization with respect to itself. See Eq. 52.

**Figure 25.** Normalized cumulative damage distribution for De Kooy weather station; $R_c = 0.07$ m, $n = 1.1$ and $V_\infty = 90$ m s$^{-1}$; no slowdown: ——; spherical droplets: $\beta = 5.7$: - - -, $\beta = 7$: ·······, $\beta = 9.58$: ——; oblate droplets: $\beta = 5.7$: - - -, $\beta = 7$: ·······, $\beta = 9.58$: ——.





The shift in "production" of the erosion damage could also influence the viability of erosion mitigation strategies such as the erosion-safe mode. The erosion-safe mode aims at avoiding damage by either reducing the tip speed or shutting down the turbine during precipitation events. To develop this point further, the damage of Eq. 52 can be expressed as $1 - \overline{\mathrm{PMD}_{\text{cumulative}}}(\beta)$ and be plotted against

$$(1 - F_I) \cdot 100\% = \left( 1 - \int_0^I f_I' dI' \right) \cdot 100\%, \tag{53}$$

resulting in Fig. 26. The figure should be interpreted as how much damage will be saved if X % of the highest intensity precipitation events can be avoided. As an example, the figure shows that for droplets without slowdown (——), turning off a turbine during the 20 % highest intensity precipitation events will reduce the erosion damage by 49 %. Likewise, avoiding the 50 % highest intensity rain events will save 79 % of all damage. When droplet deformation and slowdown are taken into account, this curve shifts. Depending on the damage exponents avoiding the 20 % most intense rain events now avoids 53 % to 55 % of the erosion damage. Alternatively, when moving laterally, 49 % of erosion damage can be saved when 15.9 % to 17.5 % of the highest rain intensity events are avoided. From the figure, it is also visible that the assumption of purely spherical droplets also shifts the curve. However, this shift's magnitude is fairly low compared to oblate droplets. To conclude, the deformation and slowdown effect reduces erosion damage and impacts the viability of erosion-mitigation strategies. In case the erosion-safe mode is used, neglecting the slowdown effect will yield a sub-optimal utilization by reducing power production in conditions that are not contributing the most towards erosion damage.



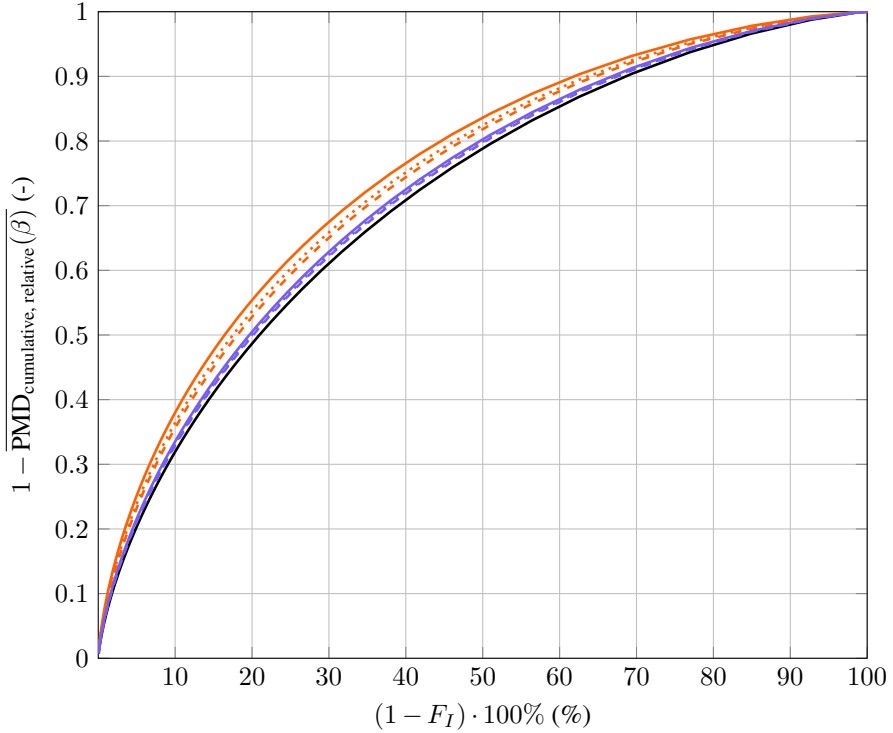

**Figure 26.** Non-dimensionalized cumulative damage distribution against the X % of heaviest rain events; $R_c = 0.07$ m, $n = 1.1$ and $V_\infty = 90$ m s$^{-1}$; no slowdown: ——; spherical droplets: $\beta = 5.7$: - - - , $\beta = 7$: ········, $\beta = 9.58$: ——; oblate droplets: $\beta = 5.7$: - - - , $\beta = 7$: ········, $\beta = 9.58$: ——.

### 3.4  Droplet behavior for model turbines

The impact of the droplet slowdown on two model turbines is investigated. As previously discussed, the NREL 5MW and IEA 15MW turbines were chosen for this purpose. The turbines were assumed to be located at the De Kooy weather station. First, the slowdown velocities are analyzed, and the resulting normalized damage distribution is subsequently investigated. Nominal turbine operating conditions at design tip speed ratio (TSR) were chosen as the control set point for the comparison (IEA TSR = 9, NREL TSR = 7.55). The parameters from Fig. 13, 14, and 15 were used for the blade elements. The ballistic

angle of attack correction coefficients of Table 4 were applied. As previously, the philosophies of the original model turbine definition were used. Meaning the airfoils of the NREL 5MW turbine stay constant between the officially defined stations, whereas linear interpolation is used between stations for the IEA 15MW turbine. Hence a saw tooth pattern in the results of the NREL turbine is expected.

Figure 27 shows the slowdown along the blades of the reference turbines. The calculations were performed for spherical and

oblate droplets and for diameters of 0.5, 1.0, 2.0, and 3.0 mm. The slowdown velocities are approximately twice as high for the IEA turbine. The reasons can be found in the slightly higher tip speed of the IEA turbine and the larger aerodynamic nose





radius, as shown in Fig. 13. The latter, as discussed in Fig. 23, is a significant driver for the slowdown of droplets. The IEA's $R_c$ is higher due to the larger chord, but also airfoils that have, in general, a higher $R_{c,0}/c$ value as shown in Table 4.

As expected, smaller droplets show a more significant slowdown along the blade. Spherical droplets experience a decrease
in the slowdown velocity from inboard to outboard. This is, at first glance, counterintuitive since the blade element speed is higher towards the tip of the blade. However, the decrease of the aerodynamic nose radius and the aerodynamic exponent, as shown in Fig. 13, offsets the increase in blade element velocity. Oblate drops show an inverted behavior where the slowdown velocities increase to a maximum when traveling outboard. There the slowdown effect starts to diminish again. As with the spherical droplets, there is a sharp drop at the blade's tip. In general, in the tip region, the slowdown velocities for oblate
droplets are about 2 to 3 times higher than for spherical droplets. Hence, the deformation of the droplets is also critical when actual wind turbines are considered. The rapidly decreasing chord can explain the sharp drop at the very tip of the blade.

The point of maximum slowdown for oblate droplets shifts outboard with decreasing droplet diameters. Larger droplets see their maximum inboard, whereas the smaller droplets see their maximum outboard of the blade. This reveals another drop size-dependent non-linearity of the slowdown effect. Larger droplets see a reduced slowdown compared to their smaller peers,
and the slowdown is unevenly distributed along the blade. Large droplets see inboard a relatively large slowdown, whereas small droplets are slowed down significantly in the erosion-prone outboard region of the blade.

An interesting observation can be made in Fig. 27d where the curves of the various droplet sizes are not only offset but also briefly overlap, e.g. at $r/R_{\text{blade}} = 0.65$ for the 0.5 and 1.0 mm diameter droplets. Even though the droplets have different sizes, they see the same absolute slowdown. This effect was found before in Fig. 22c, where a saddle point was observed. The
position of the saddle point with respect to the droplet diameter shifts for variations in $R_c$ and $n$ and thus leads to different overlapping points along the blade.

Figure 28 shows the non-dimensional damage along the blade. The damage was calculated using the Eq. 42 with the non-dimensionalization of

$$\overline{\text{PMD}(\beta)} = \frac{\text{PMD}(\beta)}{(\text{PMD}(\beta))_{\text{no slowdown}}}. \tag{54}$$

The damage was calculated for every element with $V_{\text{collection}} = V_{\text{element}}$. As before, to investigate the sensitivity of the results, the three damage exponents of 5.7, 7, and 9.58 were considered. A damage of unity represents the damage accumulated from a turbine without any droplet slowdown.

For both turbines, the damage decreases towards the blade root, which at first glance again seems counterintuitive. However, the slowdown velocities stay reasonably constant along the entire blade. In contrast, the blade section speeds vary linearly
from close to zero to 82 and 95 m s$^{-1}$ for the NREL 5MW and IEA 15MW turbines, respectively, when moving toward the blade's tip. Hence, the ratio between slowdown and blade element speed is much higher inboard of the blade, and, therefore, the slowdown leads inboard to a proportionally higher damage reduction. Still, at the blade's tip, the slowdown effect is non-negligible. Large damage reductions are observed at $r/R_{\text{blade}}$ of 0.9. Under the assumption of spherical droplets, the normalized damage is in the range of 0.8 to 0.9 for the NREL turbine. The range for oblate droplets is 0.5 to 0.7. The IEA turbine shows




slightly lower non-dimensional damages. As in Fig. 24, the band formed by the damage exponents is fairly constant along the entire blade span, indicating that the results are robust with respect to the damage exponent.

To conclude, the slowdown effect significantly impacts the lifetime prediction of actual wind turbine blades. Adding droplet deformation changes the magnitude and the characteristics of the slowdown velocity along the blade. Even though the highest damage reduction can be found inboard, the slowdown effect remains significant at the blade tip. The results of Fig. 27 and

28 show how a larger $R_c$ can effectively increase the slowdown and thus mitigate erosion damage. This lever seems especially interesting by considering the properties of the airfoils shown in Table 4, i.e. $R_{c,0}/c$ and the angle of attack correction.

(a) NREL 5MW; spherical droplets.

(b) NREL 5MW; oblate droplets.

(c) IEA 15MW; spherical droplets.

(d) IEA 15MW; oblate droplets.

**Figure 27.** Droplet slowdown along the non-dimensional blade distance of the NREL 5MW and IEA 15MW turbine; spherical and oblate droplets are considered; slowdown is shown for droplets of 0.5 mm: ———, 1.0 mm: ———, 2.0 mm: ———, 3.0 mm: ———.





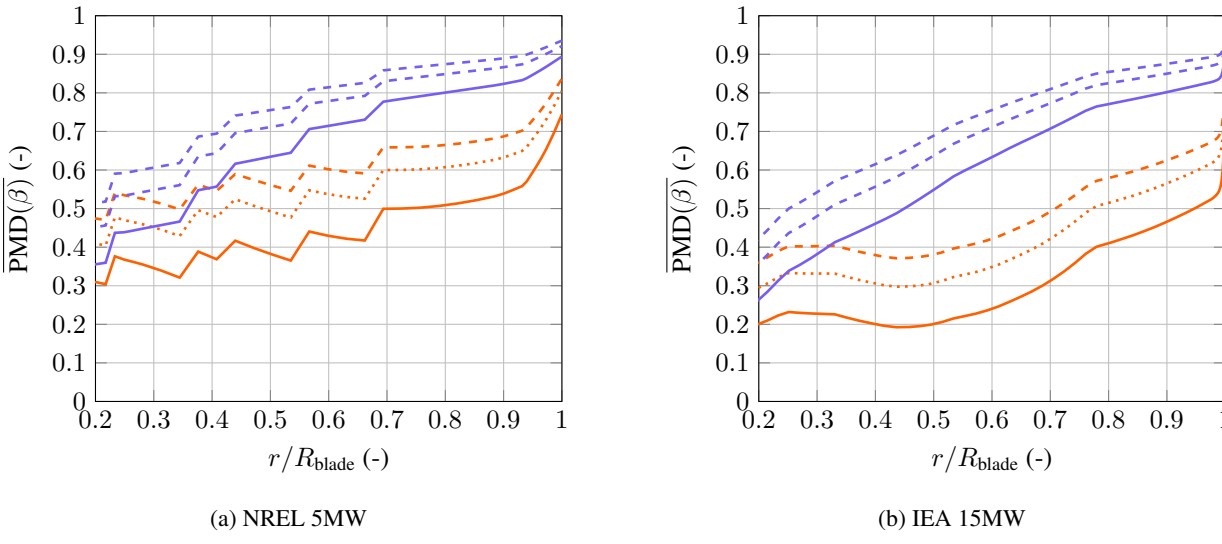

**Figure 28.** Damage distribution along the non-dimensional blade distance; spherical droplets: $\beta = 5.7$: - - -, $\beta = 7$: - - -, $\beta = 9.58$: ——; oblate droplets: $\beta = 5.7$: - - -, $\beta = 7$: ⋯⋯, $\beta = 9.58$: ——.

## 4 Conclusions

Based on previous findings in the literature, it can be said that experiments in a rotating-arm test-rig environment that used a parameter space relevant to current wind turbine designs have shown that droplets slow down and break up when they approach an airfoil. Hence, slowdown and deformation are also most likely occurring on actual wind turbines. Measurements have shown that the slowdown can be in excess of $10 \text{ m s}^{-1}$ for small droplets. The slowdown becomes less significant as the droplet diameter increases. Moreover, in the above experiments, the breakup modes of bag, bag-stamen, and shear were observed. The role of such droplet break up on rain erosion is unknown.

From the results obtained in this study, the following main conclusions can be drawn:

– The slowdown effect leads to significant damage reductions and, consequently, should not be neglected in erosion damage modeling. On actual wind turbines, the slowdown effect varies along the blade but remains significant throughout the erosion prone region. The conclusions regarding the slowdown in this work are robust with respect to variations in the model parameters, such as the exponents of the power law damage model.

– Droplet size matters! For the investigated cases, droplets under 0.25 mm diameter are slowed down so much that they contribute only marginally to the erosion damage. Large droplets are thus more damaging than their smaller peers. Furthermore, the droplet slowdown is highly sensitive towards the aerodynamics nose radius $R_c$. Due to an expected difference in trajectory between small and large droplets, the angle of attack correction of $R_{c,\alpha}$ is projected to be more significant for smaller droplets. This correction increases the slowdown of smaller droplets.



- Droplet shape matters too! The slowdown effect is already significant for spherical droplets. However, the slowdown by oblate droplets greatly exceeds that of spherical droplets. Therefore, deformation must be taken into account. When studying the impact of droplets on blades, droplets should (at least) be modeled as being oblate. Figure 22e can be used as a suggestion for a particular shape.

- Rain intensity matters! This is due to the relationship of rain intensity and droplet size distribution. The slowdown effect is particularly significant for light rain-intensity events. It also shifts the damage accumulation to higher precipitation intensities. Therefore, it may be beneficial to reduce the tip speed of turbines only during heavy precipitation events to avoid erosion.

Due to the importance of the droplet slowdown effect on the erosion lifetime of the wind turbine blades, additional research is recommended:

1. Rotating-arm erosion test-rigs might also encounter a slowdown effect. This effect would then need to be taken into account in order to find the true impact-speed for a given free-stream velocity.

2. It is conceivable that droplets might break up in a cascade decay. Additionally, it has been shown that droplets prior to impact can represent a water mass that has a non-homogeneous velocity. The potential implications on the rain erosion damage of these two effects need to be better understood.

3. In general, more research needs to be conducted on the dynamics of droplet breakup when droplets are subjected to a transient slip velocity field. For example, when approaching an airfoil. Especially the exact conditions and non-dimensional numbers that promote the various breakup modes need to be further understood. Based on such findings, a catalog of droplet shapes just prior to impact would be beneficial, as it could be used in further studies that concern the collision of droplets with wind turbine blades as well as applications beyond wind energy applications.

*Code and data availability.* The code and data can be provided on request by contacting N. Barfknecht.

*Author contributions.* N. Barfknecht conceptualized the research, developed the methodology, produced the results, and wrote the original draft. D. von Terzi supervised the research, aided with helpful discussions and reviewed and edited the paper.

*Competing interests.* The authors declare that they have no conflict of interest.



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
