# Peer review of "Aerodynamic interaction of rain and wind turbine blades: The significance of droplet slowdown and deformation for leading-edge erosion"

_Wind Energy Science, 2023_

## Author Comment (AC3)

**Manuscript ID: WES-2023-169 On the significance of rain droplet slowdown and deformation for leading-edge rain erosion**

Nils Barfknecht, Dominic von Terzi

May 2024

**Dear Reviewers,**

We would like to thank you for your time and effort in reviewing our manuscript. We believe that your comments have improved the quality of our work. Through this collaboration, we can move toward our common goal of mitigating leading-edge erosion.

Below, you can find your original comment on our work. Where appropriate, we have inserted our answers. We have spent a considerable amount of time carefully reviewing your comments. We hope that we have addressed them to your satisfaction.

Sincerely,

Nils Barfknecht and Dominic von Terzi

**Reviewer 1**

The paper addresses Leading edge erosion (LEER) of wind turbine blades (WTB). In particular it explores the relative impact velocity between droplet and WTB leading edge as function of blade velocity, droplet size, droplet shape distortion and airfoil geometry. This is of broad international interest to WT operators, manufacturers and academia. It´s a highly relevant scientific topic within the scope of WES. The topic is important for understanding some of the governing parameters of leading edge erosion and the translation from rain erosion test results to field conditions and expected blade life with respect to erosion. The objectives of the paper are clear, and the scientific methods are clearly outlined. The assumptions and analysis seem valid. The presented results are sufficient to support the interpretations and associated discussion. The discussion is relevant and relate to the results and methods. The concept of modelling shape distortion and relative impact velocity as functions of blade velocity, droplet size, and airfoil geometry ad correlating it to erosion damage is new for WTB erosion and thus is the presented data. The authors refer and give credit to earlier related work and clearly indicate their own contribution.

**Comment** The title reflects the contents of the paper. However, it will be more informative if it relates "droplet slowdown and deformation" with "aerodynamic interaction with wind turbine blade airfoil" or something like it.

We agree with the suggestions of the reviewer. We have changed the old title:
On the significance of rain droplet slowdown and deformation for leading-edge rain erosion
to
Aerodynamic interaction of rain and wind turbine blades: The significance of droplet slowdown and deformation for leading-edge erosion

The paper is generally well structured and written in good concise language. The abstract provides a concise and complete summary, including quantitative results. Figures and tables are useful and well explained in the text.

**Comment** Mathematical formulae, symbols, abbreviations, and units are appropriate. However, a list of symbols is highly recommended.

After contacting the editorial team of WES it was determined that a list of symbols can be added as an appendix. We did so.

**Comment** The methodology section contains a lot of background information and references to literature and earlier studies. This should be moved to the introduction. It could be in a section 1.1 "Background. The methodology should explain own work but can refer to the introduction and background section.

We agree that the subsection *Discussion of the underlying physics* in the section *Methodology* could be placed in a better location. While it serves as an important introduction to the Methodology, it also contains information that can be regarded as stand-alone. We, therefore, decided to have it as an independent section preceding the Methodology.

**Comment** A discussion section would be welcome. Discuss your methods. E.g. discuss how the new knowledge affects the models for expected WT blade life (with respect to erosion) based on data from rain erosion test. Velocities, nose radii and droplet sizes may be different in the two situations.

The role of the slowdown effect on rain erosion test results is, e.g., discussed in *Section 3.2 The influence of the aerodynamic nose radius on the impact velocity*. However, a full discussion is beyond the scope of the paper and would constitute an interesting follow-up project.

The number and quality of references are appropriate.

There is no supplementary material. The authors state that code and data are available upon request.

**Comment** A list of contents would enhance overview.

After contacting the editorial team of WES, it was determined that adding a list of contents is not permissible.

SPECIFIC COMMENTS

**Comment** L2: up to a few percent.

We agree. Changed.

**Comment** L115: bubble starts growing: bag starts forming

We agree. Changed to: [...], as soon as the bag starts rapidly growing,[...]

**Comment** Figure 6: own data or from literature?

Own data. Added clarification in the caption.

**Comment** L180-187 and beyond. Should be in background section. Maybe just refer to original work and present results. Is it necessary to derive?.

Yes in this case the full model needs to be given and the content should not be moved into a background section. There are a couple of reasons.

- The model is new for the wind energy community and has not been used so far. It, therefore, makes sense to write it in a consistent and easy-to-implement manner. Especially because the original model is somewhat spread over 3 original papers. Additional similarities exist with the work of Schmehl 2004.

- In the original formulation, a critical error was made in the dA/da term. There is a correction that was published 6 years after the original paper. However, we only found the correction after trying to contact the authors and after numerous hours of debugging the code.

- The approach presented here reduces the original 2D formulation to a more convenient 1D formulation while maintaining sufficient accuracy. That way, the background velocity field can be conveniently described.

- A limiter is added to the maximum deformation of the oblate spheroid. To understand this limiter, the model needs to be understood first. Referring to a collection of original papers does not increase the readability of our work.

- This paper writes all formulas in a consistent manner and adds missing information, removing ambiguity. It also writes the resulting differential equations that need to be solved. After publishing the preprint, another researcher contacted us to request help implementing the model. This supports our point.

**Comment** L205: Can that many decimals be justified?

We ran a couple of tests. 4 decimals in the coefficients lead to an accuracy up to and including the second decimal in the slowdown velocity. Due to the high value of beta, it seems appropriate to compute the slowdown up to the second decimal. For example, and error in the second decimal means $(90.01/90)^10 = 1.001$. Hence, with 4 decimals, the error is below 0.01 m/s, and thus, the lifetime is calculated with an error of less than 0.1 percent.

**Comment** L209: what is the difference between "free stream velocity" and "slip velocity"?

We added more clarification in the text. The slip velocity is the difference between the local air velocity and the drop velocity. The free stream velocity is the velocity of the droplet far away from the blade. Or, in the reference frame of the droplet, the blade velocity when the blade is still far away.

**Comment** L323 and beyond: again move to background section.

After careful consideration, we believe that the content should be left in its present location.

**Comment** L390-beyond: is it necessary to derive this, or enough to refer to Verma 2021? And it should be moved to the background section.

We updated this section by altering the notation and the equations. The equations that are presented here, e.g., the various formulations of the damage, are directly used in the results section of the text and, therefore, should not be considered as background information.

With respect to the lognormal description of the rain intensity: If the definition of $f_I$ is moved to a background section, then also $f_{phi,air}$ and $f_{phi,plane}$ should be moved. However, we believe that the entire damage model should be described in one place. We are aware that the section might seem slightly verbose. However, we believe that this verbosity is required so that the results are reproducible.

**Comment** P30 and fig.23: What is a typical range of nose radii for the outboard parts of WT blades?

This is discussed in the text. See L285 to 293 and especially Figure 13.

TECHNICAL CORRECTIONS

**Comment** Figure 3 (and others?) missing statement of permission to reproduce.

Statements of permission to reproduce were added. Permission of Figure 1, photographs in Table 1, and Figure 7 was requested prior to posting the preprint by the authors who are also the copyright holders. Permission for Figure 3 was granted by CCC Rightslink after paying a fee.

**Comment** L84: shapes: shape

Corrected.

**Comment** L85: This is process: This process

Corrected.

**Comment** Table 1: image quality could be better

We agree. But we are unfortunately limited by the image quality of Garcia Magarino 2016.

**Comment** L120: all resulting droplets: each resulting droplet

Corrected.

**Comment** L137: error in the (computed) lifetime

Corrected.

**Comment** L354: The figure: Fig.19

Corrected.

**Reviewer 2**

The manuscript addresses the issue of rain-induced erosion on wind turbine blades and identifies the lack of accuracy/knowledge for currently available models and experimental data from the literature.

A considerable amount of experimental data has been collected and analyzed to develop a new model (derived from previous models) for rain drop shape and impact velocity on the blade.

The developed model is then used to analyze the impact velocities and subsequent erosion damage on two model wind turbine blades, providing insight into the damage induced at different locations along their radius.

The models are well presented and explained with references to the relevant literature.

The plots are all very clear and so is the overlay text.

The conclusions are well presented and useful for both academia and industry.

I have a few minor comments that the authors should address to improve the quality of their manuscript:

**Comment** Line 22: Starting a sentence with a variable name is not recommended (this happens several times in the manuscript, e.g. lines 162-164, line 376, etc.).

We will discuss with the editor whether this is a problem and, depending on the outcome, change the affected areas in the text.

**Comment** Line 27: "in literature" is missing

Corrected.

**Comment** Figure 2: i) "Measure vel. →specify "relative" vel if it is so ii) free-stream vel are actually airfoil vels?

Changed caption and axis labels.

**Comment** Figures in general: have you cleared the copyright issues with the various figures you take from other papers?

See comment to technical corrections of Reviewer 1.

**Comment** The use of "drop(s)" as a synonym for of droplets makes the text difficult to read (especially since you also use the verb droping). Please correct.

We changed everything to droplets, except when talking about drop-size dependency and drop-size distributions, since, for example, droplet-size dependency sounds awkward.

**Comment** Line 130: You give DNS as the miracle solution, but DNS deals with fluid dynamics, so you mean multiphase? Please explain.

Yes, we mean multifluid/multiphase DNS. So, for example, a VOF or levelset method. In the multifluid community, the term DNS is often used when talking about resolving both fluids with the full set of NS equations. We changed the text.

**Comment** Figure 9: Can you avoid putting the numbers over the dots?

We did that on purpose. The numbers help identify the source of the data point. The dots help with extracting the exact numerical value if the user wishes to do so.

**Comment** eps" - does the choice of numerical values you use significantly affect your results?

No. We ran a small test. For $R_c = 0.1$ m/s, $n = 1.2$, $U = 90$ m/s and $\phi_0 = 0.5$ mm, eps needs to be larger than 1E-5 for a change in the 4th decimal of the slowdown velocity to become visible. Hence, 1E-12, as we suggested in the text, is by wide margins on the safe side.

**Comment** Line 242: were chosen, (add comma)

Added a comma.

**Comment** Line 244: OpenFOAM, simpleFoam

We fixed the spelling.

**Comment** Streamlines: You select some streamlines in front of the airfoil using ParaView, what is the uncertainty associated with this manual procedure?

The procedure was performed (automatically) using Paraview's Python API. In particular, we used for the upstream streamlines the *StreamTracerWithCustomSource* function. The ballistic path was extracted using the *PointLineInterpolator* function. Both methods used a starting point, which was placed 1 mm in front of the airfoil. In the simulations, the airfoil had a chord of 1 m.

**Comment** Table 4: The units for C's are degrees to the power of -2 ? This is a bit unclear, maybe improve the alpha in degrees part of the caption or maybe better just write $deg^-2$ ?

Changed to $deg^{-2}$.

**Comment** 292: 'Case F' and 'Case G' or 'Case F' and 'Case G

Changed and put Case F and Case G in italics.

**Comment** Figure 13: Why are the parameters of the 5MW not smooth along r? (You address this much later in the text, but it would be nice to see it here as well, and can you justify this philosophy of not interpolating between the airfoils a bit more?)

This was explained in Line 283, but we added more clarification. We do not interpolate in the case of the NREL 5MW turbine because of its definition. The NREL 5MW is defined such that the airfoils

stay constant between reference points. On the contrary, the IEA 15MW is defined such that everything should be linearly interpolated between reference points. We are not sure why this particular choice has been made for the NREL 5MW turbine, but we decided to simply follow their philosophy.

**Comment** Line 299: $\phi_0$ is the diameter, but it would be easier if it was stated (or if there was a nomenclature at the beginning of the paper with all the variable definitions).

A nomenclature was added, we also added a definition of $\phi_0$ at that location in the text.

**Comment** Line 331: "projected area" projected onto what?

Onto the impact target. Added clarification in the text.

**Comment** Line 354: i) You've italicized E here as opposed to reading the document i) Wouldn't $7.2 <= \beta <= 10.5$ be easier to read?

Changed the E-notation to regular exponential notation. Used the suggested format to display the range of values.

**Comment** Line 369: You might want to better explain why velocity is no longer critical in this case, since you mention that it is quite influential for damage.

After some considerations we decided to remove the sentence since it was confusing.

**Comment** Line 411: You second part →The

This was fixed after we rewrote the section as a reaction to Reviewer 1.

**Comment** Line 436: the sentence is confusing, the slowdown factors vary with the diameter of the droplet, so maybe better wording.

We rewrote the entire sentence. It should be much clearer now.

**Comment** Figure 22 (a): Droplets

Good catch. We fixed it.

**Comment** Line 461-462: Consistency in number formatting between 2 and 2.0.

Again good catch. We fixed it.

**Comment** Line 467: i.e. a factor of five larger →i.e. five times larger.

Fixed.

**Comment** Line 470-473: this is hard to follow, please expend/clarify, help the reader!

We rewrote this part and hope that it is clearer now.

**Comment** Section 3.3: the title is confusing (distribution (singular) of what? damage? intensities (plural)?

We changed the title and rewrote the lead-in sentence of the section.

**Comment** Fig. 24: caption) 1 meter H of I in mm/hr is a bit confusing?

We fixed the axis labels. It should clear now.

**Comment** Lines 528-529: avoid line return between number percent signs

We changed the format to e.g $49\sim\backslash\%$. This should fix the line return.

**Comment** Lines 540-545: See my comment on fig. 13.

See answer to previous comment.

**Comment** Lines 550-554: This is hard to follow, you should better explain how this contradicts Fig. 23 and why.

We tried to make this part clearer.

**Comment** Lines 585: start with a number

We will address this line return when doing the final formatting with the editor.